# Efficient Multi-Task Reinforcement Learning via Selective Behavior Sharing

## Abstract

Multi-task Reinforcement Learning (MTRL) offers several avenues to address the issue of sample efficiency through information sharing between tasks. However, prior MTRL methods primarily exploit data and parameter sharing, overlooking the potential of sharing learned behaviors across tasks. The few existing behavior-sharing approaches falter because they directly imitate the policies from other tasks, leading to suboptimality when different tasks require different actions for the same states. To preserve optimality, we introduce a novel, generally applicable behavior-sharing formulation that can selectively leverage other task policies as the current task's behavioral policy for data collection to efficiently learn multiple tasks simultaneously. Our proposed MTRL framework estimates the shareability between task policies and incorporates them as temporally extended behaviors to collect training data. Empirically, selective behavior sharing improves sample efficiency on a wide range of manipulation, locomotion, and navigation MTRL task families and is complementary to parameter sharing. Result videos are available at https://sites.google.com/view/qmp-mtrl.

## 1 Introduction

Imagine we are simultaneously learning to solve a diverse set of tasks in a kitchen, such as cooking an egg, washing dishes, and boiling water (see Figure 1). Several behaviors are similar across these tasks, such as interacting with the same appliances (*e.g.*, the fridge or faucet) or navigating common paths across the kitchen (*e.g.*, going to the countertop). While learning a particular task, humans can effortlessly recognize behaviors that can or cannot be shared from other tasks. This enables us to *efficiently* learn multiple tasks by selectively sharing already learned behaviors when they fit.

Can we craft a framework to replicate the human ability to selectively utilize common behaviors across tasks for efficient multi-task learning? While most works in multi-task reinforcement learning (MTRL) share information between tasks via policy parameters (Vithayathil Varghese & Mahmoud, 2020) or data relabeling (Kaelbling, 1993), recent works (Teh et al., 2017; Ghosh et al., 2018) have explored *uniform* behavior-sharing by distilling all policies into one (Rusu et al., 2015). However, distillation introduces negative interference, especially when tasks require different optimal behaviors from the same state. Our goal is to propose an efficient behavior sharing approach that can be applied to a broad range of task families without assuming which tasks should share behaviors apriori.

To address this issue, we note that, unlike humans, an RL agent does not know apriori where tasks overlap or conflict, and should instead share behaviors *selectively and adaptively* as it learns. Concretely, we propose the problem of *selective behavior sharing for general task families* in MTRL. Our key insight is that selectively sharing an agent's current behaviors across tasks can be helpful to gather more informative training data, despite potential conflicts in the final policies, as shown in human learners (Tomov et al., 2021). For example, while trying to boil water in Figure 1, a household robot can benefit by exploring behaviors found rewarding in other kitchen tasks (such as going to the faucet or to the fridge), then incorporating the helpful behaviors into its own policy, instead of randomly exploring the entire kitchen.

Two key challenges arise in selectively sharing behaviors for MTRL: how to identify and incorporate shareable behaviors. To address these challenges, we propose a simple MTRL framework called Q-switch Mixture of Policies (QMP), consisting of a Q-switch for identifying shareable behaviors, which is then used to guide data collection by incorporating a mixture of policies. First, we use the

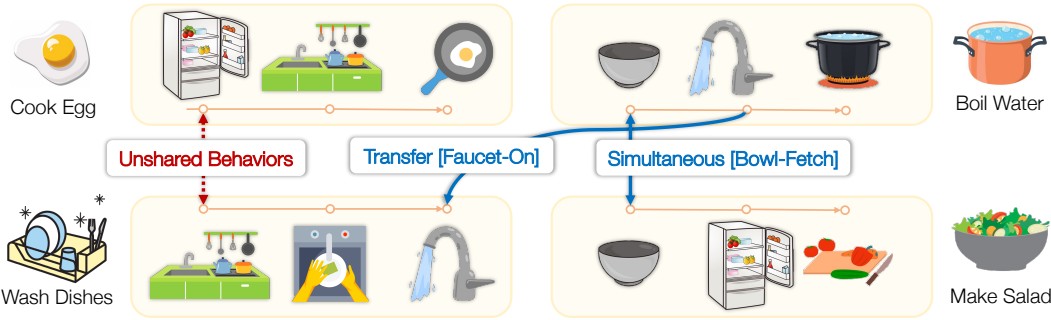

Figure 1: When an agent learns multiple tasks together, selective behavior-sharing can improve overall learning efficiency by **exploring similar behaviors simultaneously** (`[Bowl-Fetch]`) or **transferring learned behaviors** (`[Faucet-On]`) between tasks, and **avoiding unshared behaviors** (going to the refrigerator vs. sink in `Cook-Egg` and `Wash-Dishes`).

current task's Q-function (Sutton & Barto, 2018), a state and training-progress aware metric, to assess the quality of other task policies' behaviors when applied to the current task. This Q-switch acts as a filter (Nair et al., 2018) to evaluate the potential relevance of behaviors from other tasks. Second, we replace the behavioral policy used for data collection with a mixture policy that considers behavior proposals from all task policies. Specifically, the Q-switch chooses a task policy and the behavioral policy rolls it out for a fixed number of steps in order to do selective and temporally extended behavior sharing. Importantly, the mixture is only used to gather training data while each policy is still trained independently for its own task. Therefore, QMP makes no shared optimality assumptions over tasks.

Our primary contribution is introducing the problem of selective behavior sharing for MTRL in task families with conflicting behaviors. We demonstrate that our proposed framework, Q-switch Mixture of Policies (QMP), can effectively identify shareable behaviors between tasks and incorporates them to gather more informative training data. This enables sample-efficient multi-task learning in a range of manipulation, locomotion, and navigation tasks. Finally, we demonstrate QMP is complementary to parameter sharing and can be combined to yield further improvement.

## 2 RELATED WORK

**Information Sharing in Multi-Task Reinforcement Learning.** There are multiple, mostly complementary ways to share information in MTRL, *including sharing data, sharing parameters or representations, and sharing behaviors*. In offline MTRL, prior works selectively *share data* between tasks (Yu et al., 2021; 2022). Sharing parameters across policies can speed up MTRL through *shared representations* (Xu et al., 2020; D'Eramo et al., 2020; Yang et al., 2020; Sodhani et al., 2021; Misra et al., 2016; Perez et al., 2018; Devin et al., 2017; Vuorio et al., 2019; Rosenbaum et al., 2019; Yu et al., 2023; Cheng et al., 2023; Hong et al., 2022) and can be easily combined with other types of information sharing. Most similar to our work, Teh et al. (2017) and Ghosh et al. (2018) *share behaviors* between multiple policies through policy distillation and regularization. However, unlike our work, they share behavior uniformly between policies and assume that optimal behaviors are shared across tasks in most states.

**Multi-Task Learning for Diverse Task Families.** Multi-task learning in diverse task families is susceptible to *negative transfer* between dissimilar tasks that hinders training. Prior works combat this by measuring task relatedness through validation loss on tasks (Liu et al., 2022) or influence of one task to another (Fifty et al., 2021; Standley et al., 2020) to find task groupings for training. Other works focus on the challenge of multi-objective optimization (Sener & Koltun, 2018; Hessel et al., 2019; Yu et al., 2020; Liu et al., 2021; Schaul et al., 2019; Chen et al., 2018), although recent work has questioned the need for specialized methods (Kurin et al., 2022). In a similar light, we posit that prior behavior-sharing approaches for MTRL do not work well for diverse task families where different optimal behaviors are required, and thus propose to share behaviors via exploration.

**Exploration in Multi-Task Reinforcement Learning.** Our approach of modifying the behavioral policy to leverage shared task structures is related to prior work focused on MTRL exploration. Bangaru et al. (2016) proposed to encourage agents to increase their state coverage by providing an

exploration bonus. Zhang & Wang (2021) studied sharing information between agents to encourage exploration under tabular MDPs. Kalashnikov et al. (2021b) directly leverage data from policies of other specialized tasks (like grasping a ball) for their general task variant (like grasping an object). In contrast to these approaches, we do not require a pre-defined task similarity measure or exploration bonus. Skill learning can be seen as behavior sharing in a single task setting such as learning options for exploration (Machado et al., 2017; Jinnai et al., 2019b;a; Hansen et al., 2019).

**Using Q-functions as filters.** Yu et al. (2021) uses Q-functions to filter which data should be shared between tasks in a multi-task setting. In the imitation learning setting, Nair et al. (2018) and Sasaki & Yamashina (2020) use Q-functions to filter out low-quality demonstrations, so they are not used for training. In both cases, the Q-function is used to evaluate some data that can be used for training. Zhang et al. (2022) reuses pre-trained policies to learn a new task, using a Q-function as a filter to choose which pre-trained policies to regularize to as guidance. In contrast to prior works, our method uses a Q-function to evaluate explorative actions from different task policies to gather training data.

## 3 PROBLEM FORMULATION

Multi-task learning (MTL) aims to improve performance when simultaneously learning multiple related tasks by leveraging shared structures (Zhang & Yang, 2021). Multi-task reinforcement learning (MTRL) addresses sequential decision-making tasks, where an agent learns a policy to act optimally in an environment (Kaelbling et al., 1996; Wilson et al., 2007). Therefore, in addition to the typical MTL techniques, MTRL can also share *behaviors* to improve sample efficiency. However, current approaches share behaviors uniformly (Section 2) which assumes that different tasks' behaviors do not conflict with each other. To address this limitation, we seek to develop a selective behavior-sharing method that can be applied in more general task families for sample-efficient MTRL.

**Multi-Task RL with Behavior Sharing.** We aim to simultaneously learn a multi-task set $\{\mathbb{T}_1, \ldots, \mathbb{T}_T\}$ of $T$ tasks. Each task $\mathbb{T}_i$ is a Markov Decision Process (MDP) defined by state space $\mathcal{S}$, action space $\mathcal{A}$, transition probabilities $\mathcal{T}_i$, reward functions $\mathcal{R}_i$, initial state distribution $\rho_i$, and discount factor $\gamma \in \mathbb{R}$. The objective of the agent is to learn a set of $T$ policies $\{\pi_1, \ldots, \pi_T\}$, where each policy $\pi_i(a|s)$ represents the behavior on task $\mathbb{T}_i$, so as to maximize the average expected return over all tasks, where tasks are uniformly sampled during training,

$$\{\pi_1^*, \ldots, \pi_T^*\} = \max_{\{\pi_1, \ldots, \pi_T\}} \mathbb{E}_{\mathbb{T}_i \sim U(\{\mathbb{T}_1, \ldots, \mathbb{T}_T\})} \mathbb{E}_{a_t \sim \pi_i(\cdot|s_t)} \left[ \sum_{t=0}^{\infty} \gamma^t \mathcal{R}_i(s_t, a_t) \right].$$

Unlike prior works, the optimal behaviors between tasks are not assumed to necessarily coincide and can diverge at the same state, $\pi_i^*(a|s) \neq \pi_j^*(a|s)$. Thus, full-behavior sharing with reward-relabeled data (Kalashnikov et al., 2021a) or behavior regularization (Teh et al., 2017) would be suboptimal.

## 4 APPROACH

We propose to selectively share behavior from other tasks in an MTRL agent to improve the quality of the training data, which leads to two practical challenges:

- **Identifying shareable behaviors.** Behaviors from other task policies should be shared when they are potentially beneficial and avoided when known to be conflicting or irrelevant. Furthermore, when to share behaviors can depend on the state of the environment and the agent's training progress in each tasks. Therefore, we need a state and training progress-aware mechanism to evaluate behavior-sharing between each pair of tasks.

- **Incorporating shareable behaviors.** Having determined the shareable behaviors, we must effectively employ them for better learning. Without a reward relabeler, we cannot share data directly. So, we need a mechanism that can use suggestions from other task policies as a way to explore more reasonable behaviors first, in a temporally consistent manner.

### 4.1 QMP: Q-SWITCH MIXTURE OF POLICIES

For a multi-task RL agent learning from scratch, it is initially difficult to identify which behaviors are shareable. However, by estimating the most promising behaviors to share and iteratively refining this

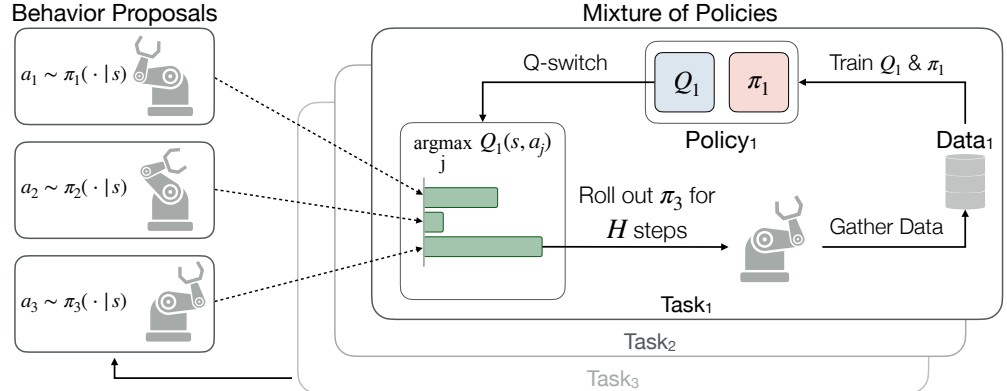

Figure 2: Our method (QMP) shares behavior between task policies in the data collection phase using a mixture of these policies. For example, in Task 1, each task policy proposes an action $a_j$. The task-specific Q-switch evaluates each $Q_1(s, a_j)$ and selects the best scored policy to gather $H$ steps of reward-labeled data to train $Q_1$ and $\pi_1$. Thus, Task 1 will be boosted by incorporating high-reward shareable behaviors into $\pi_1$ and improving $Q_1$ for subsequent Q-switch evaluations.

estimate as the agent learns the task objective, the agent can adaptively choose useful behaviors at any point in training. We propose QMP (Figure 2), a novel method that follows this intuition with two components. For each task, we use a Q-switch (Section 4.2) to evaluate the behavior proposals from a mixture of task policies (Section 4.3) which is used as the behavioral policy for data collection. We can then use any off-policy RL algorithm to update the task policy and Q-switch.

## 4.2 IDENTIFYING SHAREABLE BEHAVIORS

In MTRL, estimating sharing of behaviors from policy $\pi_j$ to $\pi_i$ depends on the task at hand $\mathbb{T}_i$, the environment state $s$, and the behavior proposal of the other policy at that state $\pi_j(s)$. Therefore, we must identify shareable behaviors in a task and state-dependent way, being aware of how all the task policies $\pi_j$ change over the course of training. For example, two task policies, such as `Boil-Water` and `Make-Salad` in Figure 1, may share only a small segment of behavior or may initially benefit from shared exploration of a common unknown environment. But eventually, their behaviors become conflicting or irrelevant to each other as the policies diverge into their own task-specific behaviors.

**Q-switch**: We propose to utilize each task's learned Q-function to evaluate shareable behaviors. The Q-function, $Q_i(s, a)$ estimates the expected discounted return of the policy after taking action $a$ at state $s$ in $\mathbb{T}_i$ (Watkins & Dayan, 1992). Although it is an estimate acquired during the course of training, it is a critical component in many state-of-the-art RL algorithms (Haarnoja et al., 2018; Lillicrap et al., 2015). It has also been used as a filter for high-quality training data (Yu et al., 2021; Nair et al., 2018; Sasaki & Yamashina, 2020), suggesting the Q-function is effective for evaluating and comparing actions during training. Thus, we use the Q-function as a switch that rates action proposals from other tasks' policies for the current task's state $s$. While the Q-function could be biased when queried with out-of-distribution actions from other policies, this is corrected in practice, as we will explain in the following section. Thus, this simple and intuitive function is state and task-dependent, gives the current best estimate of which behaviors are most helpful (those with high Q-values) and conflicting or irrelevant behaviors (those with low Q-values), and is quickly adaptive to changes in its own and other policies during online learning.

## 4.3 INCORPORATING SHAREABLE BEHAVIORS

We propose to modify only the data collection pipeline of a task, by incorporating other task policies as temporally-extended behavior proposals. This sharing scheme can be applied to any multi-task family without having to assume access to a reward re-labeler or assuming shared optimal behavior.

**Mixture of Policies**: To allow for selective behavior sharing, we use a mixture of all task policies to gather training data for each task. Training a mixture of policies is a popular approach in hierarchical RL (Çelik et al., 2021; Daniel et al., 2016; End et al., 2017; Goyal et al., 2019) to attain reusable

skills. In MTRL, we aim to benefit similarly from reusable behaviors. The main differences are that each policy is specialized to a particular task and the mixture is only used to gather training data.

We define a behavioral policy $\pi_i^{mix}(a|s)$ for each task $\mathbb{T}_i$ as a mixture over all task policies $\{\pi_j\}_{j=1}^T$. At a given state $s$, the $\pi_i^{mix}$ uses Q-switch to choose the best-scored policy $\pi_j$ based on the action proposed by that policy, $a \sim \pi_j(s)$ (Figure 2). Then, to transfer coherent, temporally extended behaviors, we roll this policy out for a fixed number of timesteps $H$ before selecting a new policy (see Appendix 13 for more details). This mixture allows us to activate multiple policies during an episode, so we can selectively incorporate behaviors in a task and state-dependent way. And while there are more complicated ways of incorporating policy proposals, such as defining a probabilistic mixture (see Appendix E.1.1), we found that our simple method QMP performs well while requiring less hyperparameter tuning and computational overhead.

---

**Algorithm 1** Q-switch Mixture of Policies (QMP)

1: **Input:** Task Set $\{\mathbb{T}_1, \ldots, \mathbb{T}_T\}$, behavior length $H$
2: Initialize policies $\{\pi_i\}_{i=1}^T$, and Q-functions $\{Q_i\}_{i=1}^T$, Data buffers $\{\mathcal{D}_i\}_{i=1}^T$
3: **for** each epoch **do**
4:     **for** $i = 1$ to $T$ **do**
5:         **while** Task $\mathbb{T}_i$ episode not terminated **do**
6:             Observe state $\mathbf{s}$
7:             **for** $j = 1$ to $T$ **do**
8:                 $\mathbf{a_j} \sim \pi_j(\mathbf{a_j}|\mathbf{s})$
9:             $j = \arg\max_j Q_i(\mathbf{s}, \mathbf{a_j})$
10:             Roll out $\pi_j$ for $H$ steps
11:             $\mathcal{D}_i \leftarrow \mathcal{D}_i \cup \{(s, \pi_j(s), r_i, s')\}_H$
12:     **for** $i = 1$ to $T$ **do**
13:         Update $\pi_i$, $Q_i$ using $\mathcal{D}_i$ with SAC for $k$ gradient steps
14: **Output:** Trained policies $\{\pi_i\}_{i=1}^T$

---

Each task policy $\pi_i$ is trained with data gathered for its own task $\mathbb{T}_i$ by the mixture of policies $\pi_i^{mix}$, thus benefiting from shared behaviors through exploration without being limited by tasks with conflicting behaviors. Altogether, QMP enables selective multi-task behavior sharing guided by the current task's Q-switch as summarized in Algorithm 1. Crucially, QMP has a *self-correcting* characteristic even when the Q-switch is training. Particularly, if the Q-switch inaccurately overestimates and selects an out-of-distribution action proposal $a_j$, then $\pi_j$ would be deployed to collect data in the environment and, in a subsequent iteration, train the Q-switch to be accurate on this action — exactly like Q-learning corrects its Q-function by exploring and making mistakes (Sutton & Barto, 2018). We additionally show that QMP converges optimally and at least as fast as SAC in Appendix G.

## 5 EXPERIMENTS

### 5.1 ENVIRONMENTS

To evaluate our proposed method, we experiment with multi-task designs in 6 manipulation, navigation, and locomotion environments, shown in Figure 3. For evaluating the effectiveness of selective behavior sharing, the complexity of these multi-task environments is determined not only by individual task difficulty, but more importantly, by the degree of similarity in behaviors between tasks. Thus to create challenging benchmarks, we ensure each task set includes tasks with either conflicting or irrelevant behavior. Further details on task setup and implementation are in Appendix Section B.

**Multistage Reacher:** The agent is tasked to solve 5 tasks of controlling a 6 DoF Jaco arm to reach multiple goals. In 4 out of the 5 tasks, the agent must reach 3 different sub-goals in order with some coinciding segments between tasks. In the 5th task, the agent's goal is to stay at its initial position for the entire episode. The observation space does not include the goal location, which must be figured out from the reward. Thus, for the same states, the 5th task directly conflicts with all the other tasks.

**Maze Navigation:** A point mass agent has to control its 2D velocity to navigate through the maze and reach the goal, where both start and goal locations are fixed in each task. The observation lacks the goal location, which should be inferred from the dense reward based on the distance to the goal. Based on the environment proposed in Fu et al. (2020), we define 10 tasks with different start and goal locations. The optimal paths for different tasks have segments that coincide and conflict.

**Meta-World Manipulation:** We use two task sets based on the Meta-World environment (Yu et al., 2019). First, **Meta-World MT10**, proposed in (Yu et al., 2019) which is a set of 10 table-top manipulation tasks involving different objects and behaviors. We also use **Meta-World CDS**, a 4-task shared-space setup proposed in Yu et al. (2021). It places the door and drawer objects next to each other on the same tabletop so that all 4 tasks (door open, door close, drawer open, drawer close) are solvable in a simultaneous multi-task setup. In both cases, the observation space consists of the

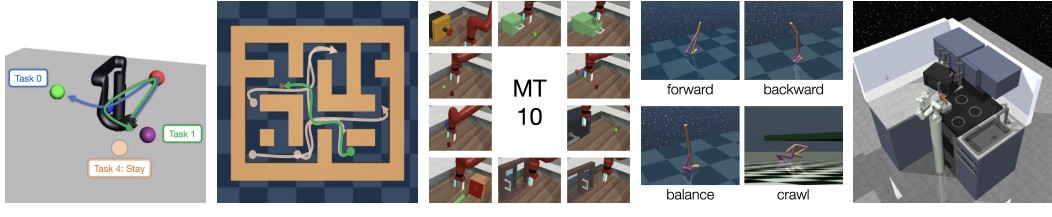

(a) Jaco Reacher    (b) Maze Navigation    (c) Meta-World    (d) Walker2D    (e) Franka Kitchen

Figure 3: **Environments & Tasks**: (a) Multistage Jaco Reacher. The agent must reach different subgoals sequentially or stay still (Task 4). (b) Maze Navigation. The agent (green circle) must navigate through the maze to reach the goal (red circle). Example paths for 4 other tasks are shown in orange. (c) Meta-World: 10 table-top manipulation tasks. (d) Walker2D: walk forward, walk backward, balance, crawl. (e) Franka Kitchen: 10 tasks, interacting with one appliance or cabinet.

robot's proprioceptive state, locations for objects present in the environment (ie. door and drawer handle for CDS, the single target object location for MT10) and the goal location. In Meta-World CDS, due to the shared environment, there are no directly conflicting task behaviors, since the policies either go to the door or the drawer, they should ignore the irrelevant behaviors of policies interacting with the other object. In Meta-World MT10, each task interacts with a different object but in an overlapping state space so there is a mix of shared and conflicting behaviors.

**Walker2D:** Walker2D is a 9 DoF bipedal walker agent with the multi-task set containing 4 locomotion tasks proposed in Lee et al. (2019): walking forward at some velocity, walking backward at some velocity, balancing under random external forces, and crawling under a ceiling. Each of these tasks involves different gaits or body positions to accomplish successfully without any obviously identifiable shared behavior in the optimal policies. In this case, behavior sharing can still be effective during training to aid exploration and share helpful intermediate behaviors, like balancing.

**Kitchen:** We use the challenging manipulation environment proposed by Gupta et al. (2019) where a 9 DoF Franka robot performs tasks in a kitchen setup. We create a task set out of 10 manipulation tasks involving turning on or off different burners and light switches, and opening or closing different cabinets. The observation space consists of the robot's state, the location of the target object, and the goal location for that object. Similar to the Meta-World CDS environment, there are no directly conflicting behaviors but plenty of irrelevant behaviors from tasks interacting with different objects.

## 5.2 BASELINES

We used Soft Actor-Critic (SAC) Haarnoja et al. (2018) for all models. We first compare different forms of cross-task behavior-sharing in isolation from other forms of information-sharing. Then, we show how behavior-sharing complements parameter-sharing. For the non-parameter sharing version, we use the same architectures and SAC hyperparameters for policies across all baselines.

- **No-Shared-Behaviors** consists of $T$ RL agents where each agent is assigned one task and trained to solve it without any behavior sharing with other agents. In every training iteration, each agent collects the data for its own task and uses it for training.

- **Fully-Shared-Behaviors** is a single SAC agent that learns one shared policy for all tasks, which outputs the same action for a given state regardless of task (thus naturally does parameter sharing too). For the fairness of comparison, we adjusted the size of the networks, batch size, and number of gradient updates to match those of other models with multiple agents.

- **Divide-and-Conquer RL (DnC)** (Ghosh et al. (2018)) uses an ensemble of $T$ policies that shares behaviors through policy distillation and regularization. We modified the method for multi-task learning by assigning each of the policies to a task and evaluating only the task-specific policy.

- **DnC (Regularization Only)** is a no policy distillation variant of DnC we propose as a baseline.

- **UDS (Data Sharing)** proposed in Yu et al. (2022) shares data between tasks, relabelling with minimum task reward. We modified this offline RL algorithm for our online set-up.

- **QMP (Ours)** learns $T$ policies sharing behaviors via Q-switch and mixture of policies.

For further details on baselines and implementation, please refer to Appendix Section F.

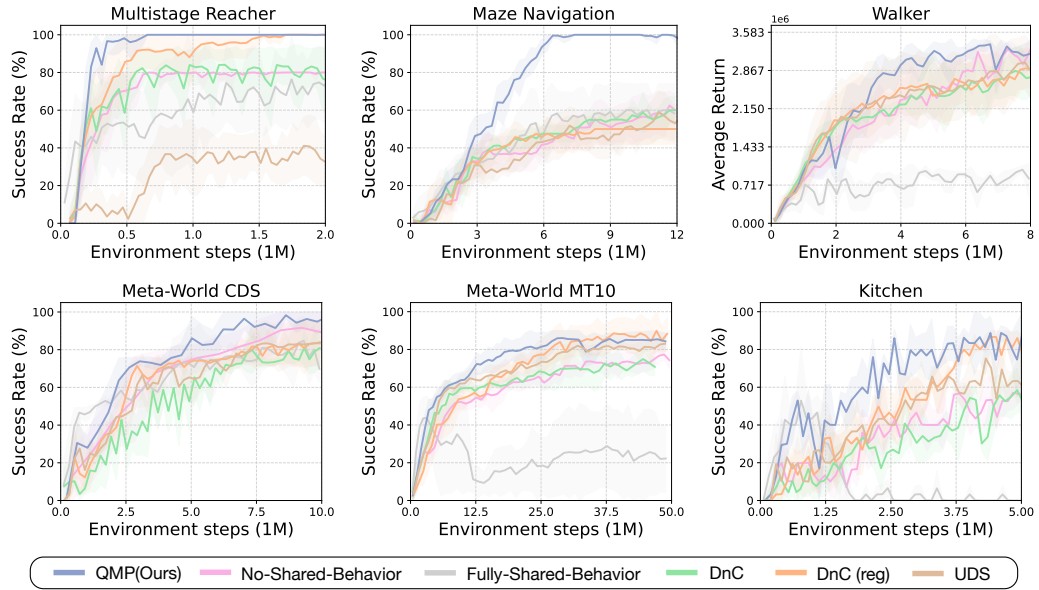

Figure 4: Comparison of average multitask success rate, over 10 evaluation episodes per task and 3-5 seeds for each method. In task sets with conflicting behaviors, QMP outperforms the baselines in terms of the rate of convergence and the task performance, and is competitive with the best-performing baseline in all other environments.

## 6 RESULTS

We conduct experiments to answer the following questions: (1) How does our method of selectively sharing behaviors for data collection compare with other forms of behavior sharing? (2) Is behavior sharing complementary to parameter sharing? (3) How crucial is adaptive behavior sharing? (4) Can QMP effectively identify shareable behaviors?

### 6.1 BASELINES: HOW DOES QMP COMPARE TO OTHER FORMS OF BEHAVIOR SHARING?

To verify QMP's efficacy as a behavior-sharing mechanism, we compare against several behavior-sharing baselines on 6 environments in Figure 4. Overall, QMP outperforms other methods in task sets with conflicting behaviors and is competitive across all task sets, demonstrating that it is an effective behavior-sharing mechanism for general task families.

In the task sets with the most directly conflicting behaviors, Multistage Reacher and Maze Navigation, our method clearly outperforms other behavior-sharing and data-sharing baselines. In Multistage Reacher, our method reaches 100% success rate at 0.5 million environment steps, while DnC (reg.), the next best method, takes 3 times the number of steps to fully converge. The rest of the methods fail to attain the maximum success rate. The UDS baseline performs particularly poorly, illustrating that data sharing can be ineffective without ground truth rewards. In the Maze Navigation environment, QMP successfully solves all tasks by 6 million environment steps, while other methods plateau at around a 60% success rate even after double the training steps.

In the remaining task sets with no directly conflicting behaviors, we see that QMP is competitive with the best-performing baseline for more complex manipulation and locomotion tasks. Particularly, in Walker2D and Meta-World CDS, we see that QMP is the most sample-efficient method and converges to a better final performance than any other multi-task RL method. This is significant because these task sets contain a majority of irrelevant behavior: between policies interacting with the door versus the drawer or the Walker in different gaits. The fact that QMP still outperforms other methods validates our hypothesis that shared behaviors can be helpful for exploration even if the optimal behaviors are different. In Meta-World MT10 and Kitchen, DnC (regularization only) also performed very well, showing that well-tuned uniform behavior sharing can be very effective in tasks without conflicting behavior. However, QMP also performs competitively and more sample efficiently,

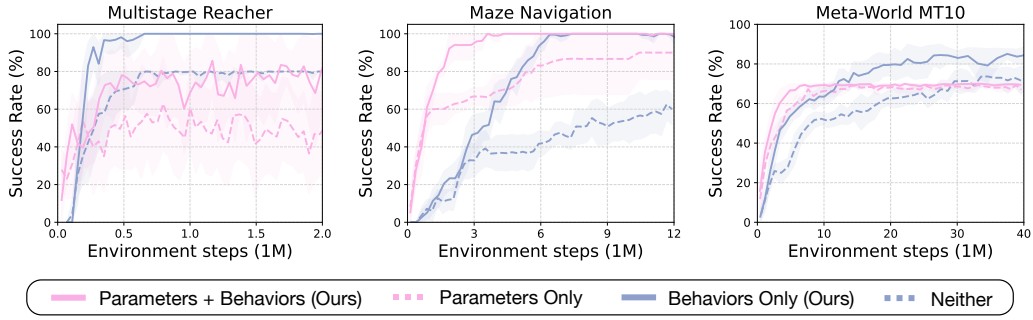

Figure 5: QMP (solid lines) improves over no behavior sharing (dashed lines) with shared parameters (pink) and no shared parameters (blue). Combining behavior sharing and parameter sharing often outperforms either method alone. Meta-World CDS and Kitchen results are in Appendix Section C.

showing that QMP is effective under the same assumptions as uniform behavior-sharing methods but can do so *adaptively* and across more *general task families*. The Fully-Shared-Behaviors baseline often performs poorly, while the No-Shared-Behavior is a surprisingly strong baseline, highlighting the challenge of behavior sharing between tasks.

## 6.2 IS BEHAVIOR SHARING COMPLEMENTARY TO PARAMETER SHARING?

It is important that our method is compatible with other forms of multitask reinforcement learning that share different kinds of information, especially parameter sharing, which is very effective under low sample regimes (Borsa et al., 2016; Sodhani et al., 2021) as we saw in the initial performance of Fully-Shared-Behaviors in Meta-World CDS. While we use completely separate policy architectures for previous experiments to isolate the effect of behavior sharing, QMP is flexible to any design where we can parameterize $T$ task-specific policies. A commonly used technique to share parameters in multi-task learning is to parameterize a single multi-task policy with a multi-head network architecture. Each head of the network outputs the action distribution for its respective task. We can easily run QMP with such a parameter-sharing multi-head network architecture by running SAC on the multi-head network and replacing the data collection policy with $\pi_i^{mix}$.

We compare the following methods in Figure 5.

- **Parameters Only**: a multi-head SAC policy sharing parameters but not behaviors over tasks.

- **Behaviors Only**: Separate task policy networks with QMP behavior sharing.

- **Parameters + Behaviors**: a multi-head SAC network sharing behaviors via QMP exploration.

Overall, we find QMP to be complementary to parameter sharing, often with additive performance gains! We find that Parameters + Behaviors generally outperforms Parameters Only, while inheriting the sample efficiency gains from parameter sharing. In many cases, the parameter-sharing methods converge sub-optimally, highlighting that shared parameter MTRL has its own challenges. However, in Maze Navigation, we find that sharing Parameters + Behaviors greatly improves the performance over both the Shared-Parameters-Only baseline *and* Shared-Behaviors-Only variant of QMP. This demonstrates the additive effect of these two forms of information sharing in MTRL. The agent initially benefits from the sample efficiency gains of the multi-head parameter-sharing architecture, while behavior sharing with QMP accelerates the exploration via the selective mixture of policies to keep learning even after the parameter-sharing effect plateaus. This result demonstrates the compatibility between QMP and parameter sharing as key ingredients to sample efficient MTRL.

## 6.3 ABLATIONS: HOW CRUCIAL IS ADAPTIVE BEHAVIOR SHARING?

We look at the importance of an adaptive, state-dependent Q-switch by comparing QMP to two ablations where we replace the Q-switch with a fixed sampling distribution over task policies to select which policy is used for exploration. **QMP-Uniform**, which replaces the Q-switch with a uniform distribution over policies, achieves only 60% success rate (Figure 6), verifying the importance of selective behavior sharing. In **QMP-Domain-Knowledge**, we replace the Q-switch with a hand-

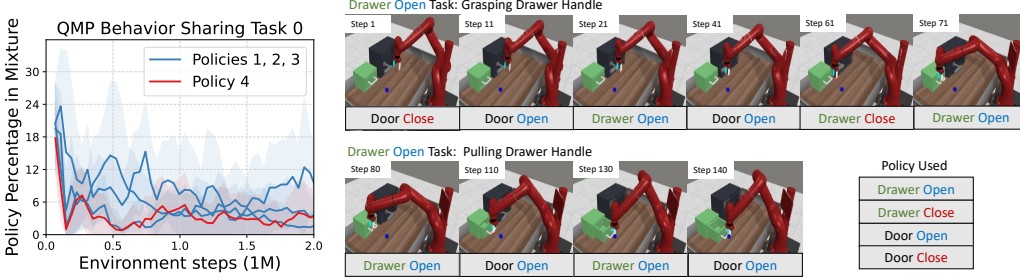

(a) Behavior-sharing over training      (b) Behavior-sharing in a single training episode.

Figure 7: (a) Mixture probabilities of other policies over the course of training for Task 0 in Multistage Reacher with the conflicting task Policy 4 shown in red. (b) Policies chosen by the QMP behavioral policy every 10 timesteps for the Drawer Open task throughout one training episode. The policy approaches and grasps the handle (top row), then pulls the drawer open (bottom row).

crafted, fixed policy distribution based on our best estimate for the similarity between tasks (i.e., sampling probabilities proportional to the number of shared sub-goal sequences between tasks in Multistage Reacher, see Appendix B for details). QMP-Domain performs well initially but plateaus early, suggesting that state-dependent and training progress-aware sharing is also necessary. Crucially, defining such a specific domain-knowledge-based mixture of policies is generally impractical and requires knowing the tasks beforehand. Additional ablations in Appendix Section E validate the gains of (i) temporal abstraction (by varying shared-behavior length $H$), (ii) arg-max mixture policy against a probabilistic mixture, and (iii) QMP's cross-task exploration against single-task SAC exploration.

## 6.4 CAN QMP EFFECTIVELY IDENTIFY SHAREABLE BEHAVIORS?

Figure 7a shows the average probability of sharing behaviors from other tasks for Multistage Reacher Task 0 over the course of training. We see that QMP learns to generally share less behavior from Policy 4 than from Policies 1-3. Conversely, QMP in Task 4 also shares the least total cross-task behavior (Appendix Figure 11). We see this same trend across all 5 Multistage Reacher tasks, showing that the Q-switch successfully identifies conflicting behaviors that should not be shared. Further, Figure 7a also shows that total behavior-sharing from other tasks goes down over training. Thus, Q-switch learns to prefer its own task-specific policy as it becomes more proficient.

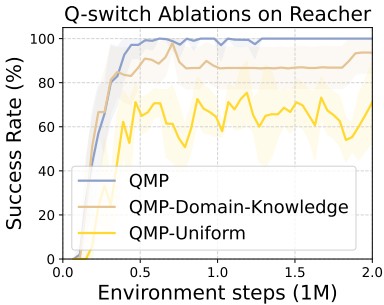

Figure 6: An adaptive state and task dependent Q-switch is crucial.

We qualitatively analyze how behavior sharing varies within a single episode by visualizing a QMP rollout during training for the Drawer Open task in Meta-World CDS (Figure 7b). We see that it makes reasonable policy choices by (i) switching between all 4 task policies as it approaches the drawer (top row), (ii) using drawer-specific policies as it grasps the drawer-handle, and (iii) using Drawer Open and Door Open policies as it pulls the drawer open (bottom row). In conjunction with the overall results, this supports our claim that QMP can effectively identify shareable behaviors between tasks. For details on this visualization and the full analysis results see Appendix Section D.

## 7 CONCLUSION

We introduce the problem of selective behavior sharing in MTRL for general task families with conflicting behaviors. We propose Q-switch Mixture of Policies (QMP), which incorporates behaviors between tasks for training data collection through a value-guided selection over behavior proposals. We demonstrate empirically that QMP effectively learns to share behavior to improve the rate of convergence and task performance in manipulation, locomotion, and navigation tasks while also being complementary to parameter sharing. Promising future directions include incorporating other forms of task knowledge, such as language instructions, to encode priors on shareable behaviors.

## 8 REPRODUCIBILITY STATEMENT

To ensure the reproducibility of our results, we have provided our code in the supplementary materials which contains all environments and also all baseline methods we report in the paper. We have also included all relevant hyperparameters and additional details on how we tuned each baseline method in Appendix Section F.

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

APPENDIX

Result videos: `https://sites.google.com/view/qmp-mtrl`

# Table of Contents

LIST OF TABLES

LIST OF FIGURES

# A   CODE SUBMISSION

In the supplementary submission, we provide the complete code to reproduce all the experiments, including QMP (ours) and baselines on all the environments.

# B   ENVIRONMENT DETAILS

## B.1   MULTISTAGE REACHER

We implement our multistage reacher tasks on top of the Open AI Gym (Brockman et al., 2016a) Reacher environment simulated in the MuJoCo physics engine (Todorov et al., 2012) by defining a sequence of 3 subgoals per task which are specified in Table 1. For all tasks, the reacher is initialized at the same start position with a small random perturbation sampled uniformly from $[-0.01, 0.01]$ for each coordinate. The observation includes the agent's proprioceptive state and how many sub-goals have been reached but not subgoal locations, as they must be inferred from the respective task's reward function.

We set up the tasks to ensure that we can evaluate behavior sharing when the task rewards are qualitatively different (see Figure 3a):

- For every task except Task 3, the reward function is the default gym reward function based on the distance to the goal, plus an additional bonus for every subgoal completed.

- For Task 1, the reward is shifted by -2 at every timestep.

- Task 3 receives only a sparse reward of 1 for every subgoal reached.

- Task 4 has one fixed goal set at its initial position.

| | Subgoal 1 | Subgoal 2 | Subgoal 3 |
|---|---|---|---|
| $T_0$ | (0.2, 0.3, 0.5) | (0.3, 0, 0.3) | (0.4, -0.3, 0.4) |
| $T_1$ | (0.2, 0.3, 0.5) | (0.3, 0, 0.3) | (0.4, 0.3, 0.2) |
| $T_2$ | (0.3, 0, 0.3) | (0.4, 0.3, 0.2) | (0.4, -0.3, 0.4) |
| $T_3$ | (0.3, 0, 0.3) | (0.4, -0.3, 0.4) | (0.2, 0.3, 0.5) |
| $T_4$ | initial | initial | initial |

Table 1: Coordinates of subgoal locations for each task in Multistage Reacher. Shared subgoals are highlighted in the same color. For Task 4, the only goal is to stay in the initial position.

**QMP-Domain**: Section 6.3 ablates the importance of an adaptive and state-dependent Q-switch by replacing it with a domain-dependent distribution over other tasks based on apparent task similarity. Specifically, to define the mixture probabilities for QMP-Domain in Multistage Reacher, we use the domain knowledge of the subgoal locations of the tasks to determine the mixture probabilities. We use the ratio of *shared sub-goal sequences* between each pair of tasks (not necessarily the shared subgoals) over the total number of sub-goal sequences, 3, to calculate how much behavior must be shared between two tasks. For that ratio of shared behavior, we distribute the probability mass uniformly between all task policies that share that behavior. For Task 4, the conflicting task, we do not do any behavior sharing and only use $\pi_4$ to gather data.

Each Task $\mathbb{T}_i$ consists of 3 sub-goal sequences $\{S_0, S_1, S_2\}$ (e.g. [initial $\rightarrow$ Subgoal 1], [Subgoal 1 $\rightarrow$ Subgoal 2], and [Subgoal 2 $\rightarrow$ Subgoal 3]). For each sequence $s \in \{S_0, S_1, S_2\}$, we divide equally the contribution of each task $\mathbb{T}_j$'s policy $\pi_j$ that shares the sequence $s$ (i.e. if $\mathbb{T}_0$ and $\mathbb{T}_1$ both contain sequence $s$, where we use the notation $\mathbb{1}(s \in \mathbb{T}_i)$ as the indicator function for whether Task $\mathbb{T}_i$ contains sequence $s$, then $\pi_0$ and $\pi_1$ both have $\frac{1}{2}$ contribution for $s$). Each sequence contributes equally to the overall mixture probabilities for Task $i$ (i.e. all policies that shares sequence $S_i$ contributes in total $\frac{1}{3}$ to the mixture probability for the behavior policy of Task $\mathbb{T}_i$). Thus, the

contribution probability of Policy $\pi_j$ to Task $\mathbb{T}_i$ is:

$$p_{j \to i} = \sum_{s \in \{S_0, S_1, S_2\}} \frac{1}{3} \cdot \frac{\mathbb{1}(s \in \mathbb{T}_j)}{\sum_k \mathbb{1}(s \in \mathbb{T}_k)}$$

$$\pi_i^{mix} = \sum_j p_{j \to i} \, \pi_j$$

Reusing notation for mixture probabilities, we have,

$$\pi_0^{mix} = \frac{2}{3}\pi_0 + \frac{1}{3}\pi_1$$

$$\pi_1^{mix} = \frac{1}{3}\pi_0 + \frac{2}{3}\pi_1$$

$$\pi_2^{mix} = \frac{5}{6}\pi_2 + \frac{1}{6}\pi_3$$

$$\pi_3^{mix} = \frac{1}{6}\pi_2 + \frac{5}{6}\pi_3$$

$$\pi_4^{mix} = \pi_4$$

## B.2 MAZE NAVIGATION

The layout and dynamics of the maze follow Fu et al. (2020), but since their original design aims to train a single agent to reach a fixed goal from multiple start locations, we modified it to have both start and goal locations fixed in each task, as in Nam et al. (2022). The start location is still perturbed with a small noise to avoid memorizing the task. The observation consists of the agent's current position and velocity. But, it lacks the goal location, which should be inferred from the dense reward based on the distance to the goal. The layout we used is LARGE_MAZE which is an $8 \times 11$ maze with paths blocked by walls. The complete set of 10 tasks is visualized in Figure 12, where green and red spots correspond to the start and goal locations, respectively. The environment provides an agent a dense reward of $\exp(-dist)$ where $dist$ is a linear distance between the agent's current position and the goal location. It also gives a penalty of 1 at each timestep in order to prevent the agent from exploiting the reward by staying near the goal. The episode terminates either as soon as the goal is reached by having $dist < 0.5$ or when 600 timesteps have passed.

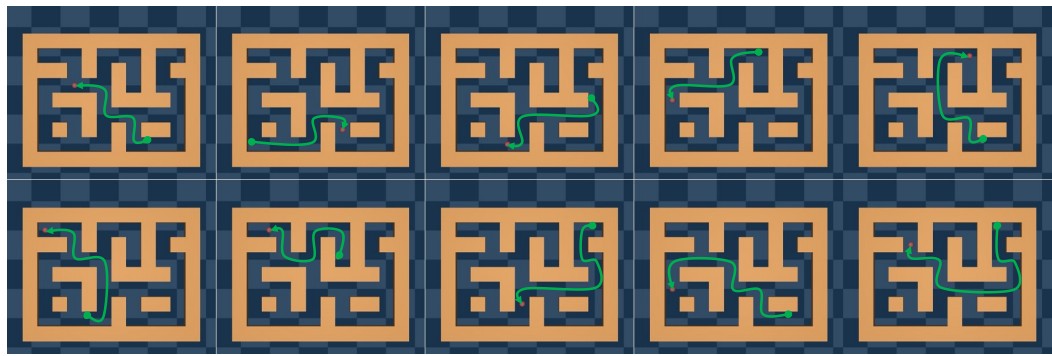

Figure 12: Ten tasks defined for the Maze Navigation. The start and goal locations in each task are shown in green and red spots, respectively, and an example path is shown in green.

## B.3 META-WORLD MANIPULATION

For Meta-World CDS, we reproduce the Meta-world environment proposed by Yu et al. (2021) using the Meta-world codebase (Yu et al., 2019), where the door and drawer are both placed side-by-side on the tabletop for all tasks (see Figure 3c). The observation space consists of the robot's proprioceptive state, the drawer handle state, the door handle state, and the goal location, which

varies based on the task. Unlike Yu et al. (2021), we additionally remove the previous state from the observation space so the policies cannot easily infer the current task, making it a challenging multi-task environment. The environment also uses the default Meta-World reward functions which is composed of two distance-based rewards: distance between the agent end effector and the object, and distance between the object and its goal location. We use this modified environment instead of the Meta-world benchmark because our problem formulation of simultaneous multi-task RL requires a consistent environment across tasks. For Meta-World MT10, we directly use the implementation provided in (Yu et al., 2019) without changes.

### B.4 WALKER2D

Our Walker2D task set uses 4 tasks proposed and implemented by Lee et al. (2019): walking forward at a target velocity, walking backward at a target velocity, balancing under random external forces, and crawling under a ceiling. Each of these tasks involves different gaits or body positions to accomplish successfully without any obviously identifiable shared behavior in the optimal policies. Behavior sharing can still be effective during training to aid exploration and share helpful intermediate behaviors, like balancing. However, there is no obviously identifiable conflicting behavior either in this task set. Because each task requires a different gait, it is unlikely for states to recur between tasks and even less likely for states that are shared to require conflicting behaviors. For instance, it is common for all policies to struggle and fall at the beginning of training, but all tasks would require similar stabilizing and correcting behavior over these states.

### B.5 KITCHEN

We modify the Franka Kitchen environment proposed by Gupta et al. (2019) and based on the implementation from Fu et al. (2020). Since this environment is typically used for long horizon or offline RL, we chose shorter tasks that are learnable with online RL. Furthermore, we added a dense reward based on the Meta-World reward function. We formed our 10 task MTRL set by choosing 10 available tasks in the kitchen environment that interacted with the same objects: turning the top burner on or off, turning the bottom burner on or off, turning the light switch on and off, open or closing the sliding cabinet, and opening and closing the hinge cabinet. Similar to the Meta-World CDS environment, these tasks may share behaviors navigating around the kitchen to the target object but have plenty of irrelevant behavior between tasks that interact with different objects and conflicting behaviors when opening or closing the same object.

## C ADDITIONAL RESULTS

### C.1 MULTISTAGE REACHER PER TASK RESULTS

Additional results and analysis on Multistage Reacher are shown in Figure 8. QMP outperforms all the baselines in this task set, as shown in Figure 4. Task 3 receives only a sparse reward and, thus, can benefit the most from shared exploration. We observe that QMP gains the most performance boost due to selective behavior-sharing in Task 3. The No-Shared-Behavior baseline is unable to solve Task 3 at all due to its sparse reward nature. The other baselines which share uniformly suffer at Task 3, likely because they also share behaviors from other conflicting tasks, especially Task 4. We explore this further in the following Section D.

For all tasks, QMP outperforms or is comparable to No-Shared-Behavior, which shows that selective behavior-sharing can help accelerate learning when task behaviors are shareable and is robust when tasks conflict. Fully-Shared-Behavior especially underperforms in Tasks 2 and 3, which require conflicting behaviors upon reaching Subgoal 1, as defined in Table 1. In contrast, it excels at the beginning of Task 0 and Task 1 as their required behaviors are completely shared. However, it suffers after Subgoal 2, as the task objectives diverge.

### C.2 ADDITIONAL QMP + PARAMETER SHARING RESULTS

We report the parameter sharing results for Meta-World CDS (Figure 9a) and Kitchen (Figure 9b). Adding behavior sharing on top of parameter sharing in these environments does not harm perfor-

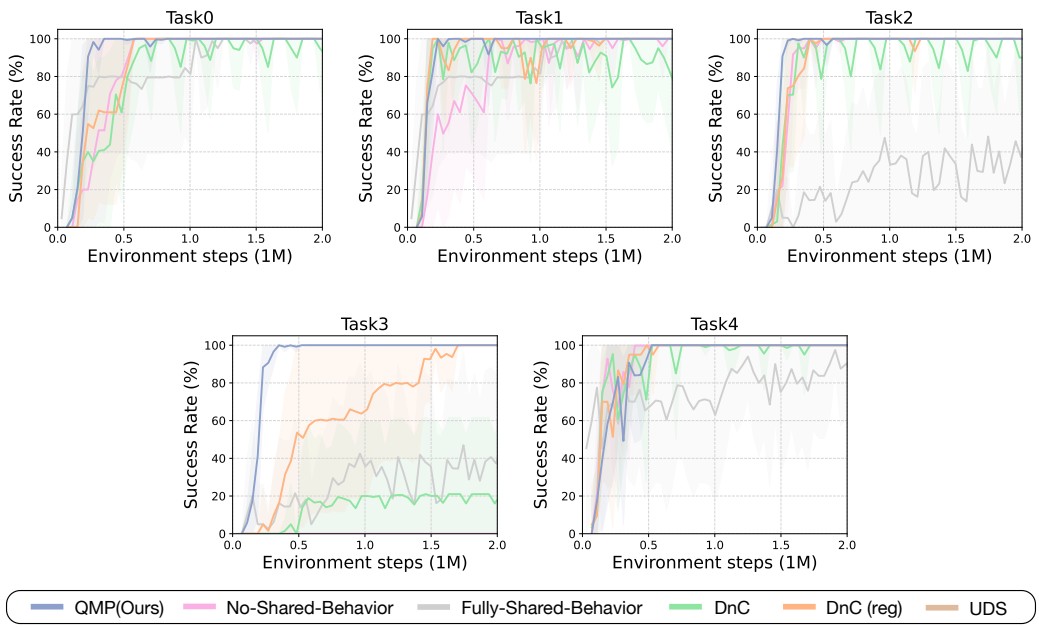

Figure 8: Success rates for individual tasks in Multistage Reacher. Our method especially helps in learning Task 3, which requires extra exploration because it only receives a sparse reward.

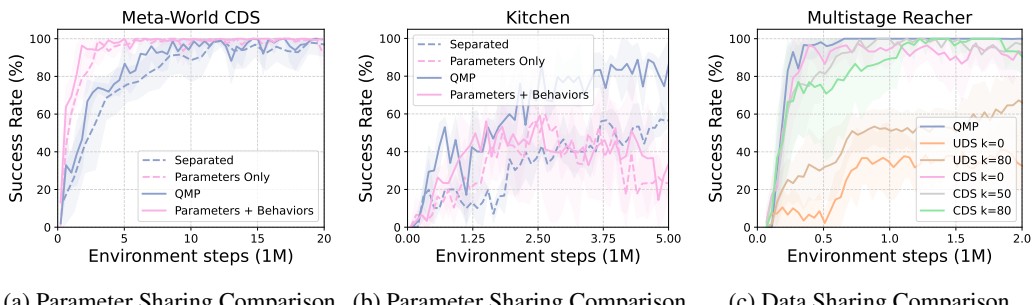

(a) Parameter Sharing Comparison   (b) Parameter Sharing Comparison   (c) Data Sharing Comparison

Figure 9: Combining our method with parameter sharing in Meta-World CDS (a) and Kitchen (b). (c) Online data sharing is very efficient when given task reward functions (all CDS versions), but suffers without (all UDS versions).

mance and suggests that both types of information sharing are necessary components in a universal MTRL method. We also note that while parameter sharing performs well in Meta-World CDS, it is difficult to optimize in the Kitchen environment and actually does worse than no parameter sharing, which likely also hampers the effectiveness of Behavior + Parameter Sharing in these tasks.

## C.3   DATA SHARING RESULTS

In Figure 9c, we report multiple sharing percentiles for UDS and for CDS (Yu et al., 2021) which assumes access to ground truth task reward functions which it uses to re-label the shared data. When the shared data is relabeled with task reward functions, thereby bypassing the conflicting behavior problem, online data sharing approaches can work very well. But when unsupervised, we see that online data sharing can actually harm performance in environments with conflicting tasks, with the more conservative data sharing approach (UDS k=80) out-performing sharing all data. $k$ is the percentile above with we share a transition between tasks, with higher $k$ representing more conservative data sharing. Full details on our online UDS and CDS implementation are in Section F.6 .

## D   QMP Behavior Sharing Analysis

**QMP learns to not share from conflicting tasks**: We visualize the mixture probabilities per task of other policies in Figure 11 for Multistage Reacher, highlighting the conflicting Task 4 in red. Throughout the training, we see that QMP learns to generally share less behavior from Policy 4 than other policies in Tasks 0-3 and shares the least total cross-task behavior in Task 4. This supports our claim that the Q-switch can identify conflicting behaviors that should not be shared. We also note that Task 3 has a relatively larger amount of sharing than other tasks. The sparse reward nature of Task 3 makes it benefit the most from exploration via selective behavior-sharing from other tasks.

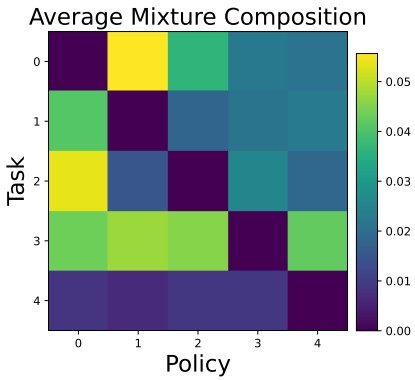

Figure 10 analyzes the effectiveness of the Q-switch in identifying shareable behaviors by visualizing the average proportion that each task policy is selected for another task over the course of training in the Multistage Reacher environment. This average mixture composition statistic intuitively analyzes whether QMP iden-

Figure 10: Proportion of shared behavior on Reacher Multistage averaged over training: Each cell (row $i$, col $j$) represents sharing contribution of Policy $j$ for Task $i$ (diagonal zeroed out for contrast).

tifies shareable behaviors between similar tasks and avoids behavior sharing between conflicting or irrelevant tasks. As we expect, the Q-switch for Task 4 utilizes the least behavior from other policies (bottom row), and Policy 4 shares the least with other tasks (rightmost column). Since the agent at Task 4 is rewarded to stay at its initial position, this behavior conflicts with all the other goal-reaching tasks. Of the remaining tasks, Task 0 and 1 share the most similar goal sequence, so it is intuitive why they benefit from shared exploration and are often selected by their respective Q-switches. Finally, unlike the other tasks, Task 3 receives only a sparse reward and therefore relies heavily on shared exploration. In fact, QMP demonstrates the greatest advantage in this task (Appendix Figure 8).

**Behavior-sharing reduces over training**: Figure 11 shows that the total amount of behavior-sharing decreases over the course of training in all tasks, which demonstrates a naturally arising preference in the Q-switch for the task-specific policy as it becomes more proficient at its own task.

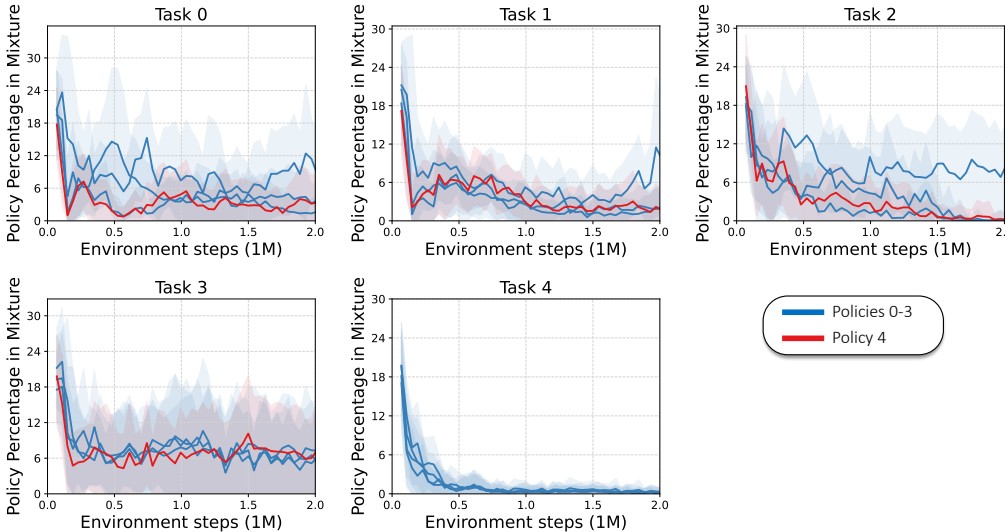

Figure 11: Mixture probabilities per task of other policies over the course of training for Multistage Reacher. The conflicting task Policy 4, which requires staying stationary, is highlighted in red.

### D.1 QUALITATIVE VISUALIZATION OF BEHAVIOR-SHARING

We qualitatively analyze behavior sharing by visualizing a rollout of QMP during training for the Drawer Open task in Meta-World Manipulation (Figure 7b). To generate this visualization, we use a QMP rollout during training before the policy converges to see how behaviors are shared and aid learning. For clarity, we first subsample the episodes timesteps by 10 and only report timesteps when the activated policy changes to a new one (ie. from timestep 80 to 110, QMP chose the Drawer Open policy). We qualitatively break down the episode into when the agent is approaching the drawer (top row; Steps 1-60), grasping the handle (top row; Steps 61-80), and pulling the drawer open (bottom row). This allows us to see that it switches between all task policies as it approaches the drawer, uses drawer-specific policies as it grasps the handle, and opening-specific policies as it pulls the drawer open. This suggests that in addition to ignoring conflicting behaviors, QMP is able to identify helpful behaviors to share. We note that QMP is not perfect at policy selection throughout the entire rollout, and it is also hard to interpret these shared behaviors exactly because the policies themselves are only partially trained, as this rollout is from the middle of training. However, in conjunction with the overall results and analysis, this supports our claim that QMP can effectively identify shareable behaviors between tasks.

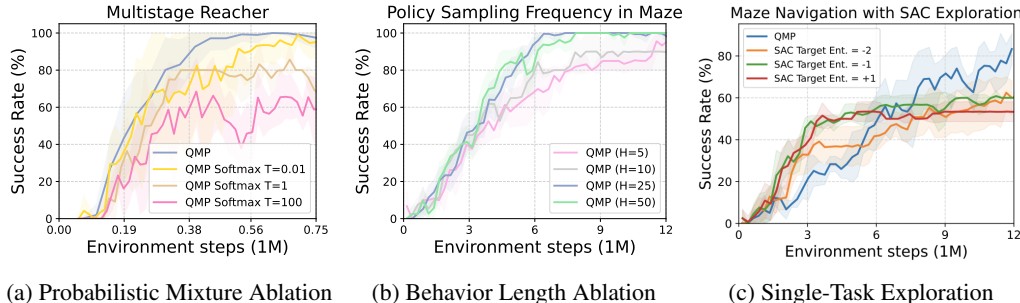

(a) Probabilistic Mixture Ablation    (b) Behavior Length Ablation    (c) Single-Task Exploration

Figure 12: (a) Using probabilistic mixtures with QMP by using a softmax over Q values with temperature T, which determines the spread of the distribution. (b) Sharing more temporally extended behaviors improves QMP effectiveness in Maze Navigation. (c) Single-task exploration by varying SAC target entropy. QMP reaches a higher success rate because it shares exploratory behavior **across** tasks.

## E ADDITIONAL ABLATIONS AND ANALYSIS

### E.1 QMP BEHAVIOR LENGTH ABLATION

We ablate the effect of different policy behavior lengths in Maze Navigation in Figure 12b, by rolling out the chosen task policy for 5, 10, 25, and 50 timesteps. We see that sample efficiency and final performance improves when sharing longer behaviors (25 and 50 timesteps), since the tasks can share more coherent, temporally extended behaviors rather than shorter action sequences. This is true despite the fact that we choose the behavioral policy greedily, only evaluating at the current state $s$. The optimal policy behavior length varies across environments since it is dependent on the task horizon and environment step frequency, and was the main hyperparameter that required tuning for QMP.

### E.1.1 PROBABILISTIC MIXTURE V/S ARG-MAX

A probabilistic mixture of policies is a design choice of our approach where arg-max is replaced with softmax. However, in our initial experiments, we found no significant improvement in performance and it came with an additional hyperparameter of tuning the temperature coefficient. As we see in Figure 12a, QMP actually outperforms a probabilistic mixture over a range of softmax temperatures, justifying the design choice of argmax over softmax due to its reliable performance and simplicity.

### E.2 QMP v/s Increasing Single Task Exploration

Since QMP seeks to gather more informative training data for the task by modifying the behavioral policy, it can be viewed as a form of multi-task exploration. We briefly investigate how single task exploration differs from multi-task exploration by tuning the target entropy in SAC in Figure 12c which influences the policy entropy and therefore exploration. We see that while tuning this exploration parameter affects the sample efficiency by more quickly learning each individual task, QMP achieves a higher final success rate by incorporating behaviors form other tasks, and therefore doing multi-task exploration. The benefit of exploration or behavior sharing algorithms specialized for multi-task RL is precisely this ability to transfer and share information between tasks.

## F Implementation Details

The SAC implementation we used in all our experiments is based on the open-source implementation from Garage (garage contributors, 2019). We used fully connected layers for the policies and Q-functions with the default hyperparameters listed in Table 2. For DnC baselines, we reproduced the method in Garage to the best of our ability with minimal modifications.

We used PyTorch (Paszke et al., 2019) for our implementation. We run the experiments primarily on machines with either NVIDIA GeForce RTX 2080 Ti or RTX 3090. Most experiments take around one day or less on an RTX 3090 to run. We use the Weights & Biases tool (Biewald, 2020) for logging and tracking experiments. All the environments were developed using the OpenAI Gym interface (Brockman et al., 2016b).

### F.1 Hyperparameters

Table 2 details the list of important hyperparameters on all the 3 environments. For all environments, we used a 2 layer fully connected network with hidden dimension 256 and a tanh activation function for the policies and Q functions. We use a target network for the Q function with target update $\tau = 0.995$ and trained with an RL discount of $\gamma = 0.99$.

Table 2: QMP hyperparameters.

| Hyperparameter | Multistage Reacher | Maze Navigation | Meta-World CDS |
|---|---|---|---|
| Behavior Length $H$ | 10 | 25 | 10 |
| Minimum buffer size (per task) | 10000 | 3000 | 10000 |
| # Environment steps per update (per task) | 1000 | 600 | 500 |
| # Gradient steps per update (per task) | 100 | 100 | 50 |
| Batch size | 32 | 256 | 256 |
| Learning rates for $\pi$, $Q$ and $\alpha$ | 0.0003 | 0.0003 | 0.0015 |

| Hyperparameter | Meta-World MT10 | Walker | Kitchen |
|---|---|---|---|
| Behavior Length $H$ | 10 | 1 | 10 |
| Minimum buffer size (per task) | 500 | 2500 | 200 |
| # Environment steps per update (per task) | 500 | 1000 | 200 |
| # Gradient steps per update (per task) | 50 | 1500 | 50 |
| Batch size | 2560 | 256 | 1280 |
| Learning rates for $\pi$, $Q$ and $\alpha$ | 0.0015 | 0.0003 | 0.0003 |

### F.2 No-Shared-Behaviors

All $T$ networks have the same architecture with the hyperparameters presented in Table 2.

### F.3 FULLY-SHARED-BEHAVIORS

Since it is the only model with a single policy, we increased the number of parameters in the network to match others and tuned the learning rate. The hidden dimension of each layer is 600 in Multistage Reacher, 834 in Maze Navigation, and 512 in Meta-World Manipulation, and we kept the number of layers at 2. The number of environment steps as well as the number of gradient steps per update were increased by $T$ times so that the total number of steps could match those in other models. For the learning rate, we tried 4 different values (0.0003, 0.0005, 0.001, 0.0015) and chose the most performant one. The actual learning rate used for each experiment is 0.0003 in Multistage Reacher and Maze Navigation, and 0.001 in Meta-World Manipulation.

This modification also applies to the Shared Multihead baseline, but with separate tuning for the network size and learning rates. In Multistage Reacher, we used layers with hidden dimensions of 512 and 0.001 as the final learning rate. In Maze Navigation, we used 834 for hidden dimensions and 0.0003 for the learning rate.

### F.4 DNC

We used the same hyperparameters as in Separated, while the policy distillation parameters and the regularization coefficients were manually tuned. Following the settings in the original DnC (Ghosh et al., 2018), we adjusted the period of policy distillation to have 10 distillations over the course of training. The number of distillation epochs was set to 500 to ensure that the distillation is completed. The regularization coefficients were searched among 5 values (0.0001, 0.001, 0.01, 0.1, 1), and we chose the best one. Note that this search was done separately for DnC and DnC with regularization only. For DnC, the coefficients we used are: 0.001 in Multistage Reacher and Maze Navigation, and 0.001 in Meta-World Manipulation. For Dnc with regularization only, the values are: 0.001 in Multistage Reacher, 0.0001 in Maze Navigation, and 0.001 in Meta-World Manipulation.

### F.5 QMP (OURS)

Our method also uses the default hyperparameters. We experimented with an optional 'mixture warmup period' hyperparameter to decide when to start using the mixture of policies in exploration. Before warmup, each agent collects data using its own policy as an exploration policy. We searched over 3 values: 0, 50, or 100 iterations. We found the option of 0 warmup iterations to perform the best across all the environments. We also tried different policy behavior lengths for each environment: 1, 5, 10, 25, and found that this was typically the only QMP hyperparameter that needed to be tuned. The exception is Meta-World MT10, where we found it helpful to have more conservative behavior sharing by choosing the task-specific policy 70% of the time. The remaining 30% we use the Q-filter to select a policy as usual.

Like in Baseline Multihead (Parameters-Only), the QMP Multihead architecture (Parameters+Behaviors) also required a separate tuning. Since QMP Multihead effectively has one network, we increased the network size in accordance with Baseline Multihead and tuned the learning rate in addition to the mixture warmup period. The best-performing combinations of these parameters we found are 0 and 0.001 in Multistage Reacher, and 100 and 0.0003 in Maze Navigation, respectively.

### F.6 ONLINE UDS

Yu et al. (2022) proposes an offline multi-task RL method (UDS) that shares data between tasks if their conservative Q value falls above the $k^{th}$ percentile of the task data. Specifically, before training, you would go through all the tasks' data and share some data from Task $j$ to Task $i$ if the Task $i$ Q value of that data is greater than $k\%$ of the Q values of Task $i$'s data. UDS does not require access to task reward functions like other data-sharing approaches. It simply re-labels any shared data with the minimum task reward, making it applicable to our problem setting as we also do not assume that reward relabeling is possible.

In order to adapt UDS to online RL, instead of doing data sharing once on the given multi-task dataset, we apply UDS data sharing before every training iteration to the data in the multi-task replay buffers. Concretely, we implement this on-the-fly for every batch of sampled data by sampling one batch of data from Task $i$'s replay buffer, $\beta_i$, and one batch of data from the other task's replay buffers $\beta_{j \neq i}$.

Then following UDS, we would form the effective batch $\beta_i^{\text{eff}}$ by sharing data from $\beta_{j \neq i}$ if it falls above the $k^{th}$ percentile of Q values for $\beta_i$:

$$UDS_{\text{online}} : (s, a, r_i, s') \sim \beta_{j \neq i} \in \beta_i^{\text{eff}}$$

$$\text{if } \Delta^{\pi}(s, a) := \hat{Q}^{\pi}(s, a, i) - P_{k^{\text{th}}}[\hat{Q}^{\pi}(s', a', i) : s', a' \sim \beta_i] \geq 0$$

Note the differences here: (i) the 'data' used for data-sharing is the sampled replay buffer batch instead of the offline dataset, and (ii) we use the standard Q-function to evaluate data instead of the conservative Q-function since we are doing online (not offline) RL. We implement it this way as a practical approximation to avoid having to process the entire replay buffer every training iteration.

We use the same default hyperparameters as the other baseline methods. Additionally, we need to tune the sharing percentile $k$. For this, we tried $0^{\text{th}}$ percentile (sharing all data) and $80^{\text{th}}$ percentile, and chose the best-performing one.

## G  QMP CONVERGENCE GUARANTEES

We derive the convergence guarantees for *Soft Mixture Policy Iteration* used in the QMP Algorithm 1 for $H = 1$, i.e., without temporally extended behavior sharing. We augment the derivation of Soft Policy Iteration in SAC (Haarnoja et al., 2018), which is our base algorithm, with our proposed QMP's mixture policy. Like SAC, we consider the tabular setting and show that QMP's modification to soft policy iteration converges to the optimal policy. The derivation sketch follows:

1. Soft *Mixture* Policy Evaluation: Since QMP modifies the soft Bellman backup operator by acting and collecting data with the mixture policy $\pi_i^{mix}$, we need to prove that the resulting policy evaluation step converges to the soft Q-value. To do this, we show that the modified soft mixture Bellman operator is a contraction in Theorem G.1.

2. Soft Policy Improvement: Since QMP does NOT modify the SAC update procedure, we can directly use SAC's guarantees of policy improvement following Lemma 2 from Haarnoja et al. (2018).

3. Soft *Mixture* Policy Improvement: We demonstrate QMP's mixture policy $\pi_i^{mix}$ guarantees to be no worse than the per-task policies $\pi_i$ that compose the mixture. In Theorem G.2, we show this by proving that the expected return following $\pi_i^{mix}$ is a weak monotonic improvement over the soft Q-value learned with $\pi_i$.

4. Soft *Mixture* Policy Iteration: In Theorem G.3, we show that the repeated application of the above 3 steps in QMP converges to an optimal policy for each task. Furthermore, the convergence rate is faster due to an enhancement of policy improvement due to Soft *Mixture* Policy Improvement.

**Theorem G.1** (Soft Mixture Policy Evaluation). *Let $\pi_i$ and $Q_i$ be the standard per-task policy and per-task Q-function for each task $\mathbb{T}_i$. Consider the mixture policy $\pi_i^{mix}$ and a soft mixture Bellman backup operator $\mathcal{T}^{\pi_i^{mix}}$ be*

$$\mathcal{T}^{\pi_i^{mix}} Q_i(s_t, a_t) \triangleq r_i(s_t, a_t) + \gamma \mathbb{E}_{s_{t+1} \sim p}[V_i(s_{t+1})], \tag{1}$$

*where*

$$V_i(s_t) = \mathbb{E}_{a_t \sim \pi_i^{mix}}[Q_i(s_t, a_t) - \log \pi_i(a_t|s_t)] \tag{2}$$

*is the soft state value function. Consider an initial mapping $Q_i^0 : S \times A \to \mathbb{R}$ with $|A| < \infty$, and define $Q_i^{k+1} = \mathcal{T}^{\pi_i^{mix}} Q_i^k$. Then the sequence $Q_i^k$ will converge as $k \to \infty$.*

*Proof.* We define the entropy-augmented per-task reward as:

$$r_{\pi_i}(s_t, a_t) \triangleq r_i(s_t, a_t) + \mathbb{E}_{s_{t+1} \sim p}[\mathcal{H}(\pi_i(\cdot|s_{t+1}))], \tag{3}$$

and rewrite the update rule as

$$Q_i(s_t, a_t) \leftarrow r_{\pi_i}(s_t, a_t) + \gamma \mathbb{E}_{s_{t+1} \sim p, a_{t+1} \sim \pi_i^{mix}}[Q_i(s_{t+1}, a_{t+1})]. \tag{4}$$

Below we show that the soft mixture Bellman operator $\mathcal{T}^{\pi_i^{mix}}$, defined as

$$\mathcal{T}^{\pi_i^{mix}} Q_i(s_t, a_t) \triangleq r_{\pi_i}(s_t, a_t) + \gamma \mathbb{E}_{s_{t+1}\sim p, a_{t+1}\sim \pi_i^{mix}}[Q_i(s_{t+1}, a_{t+1})] \tag{5}$$

is a contraction. Then similar to Haarnoja et al. (2018) (Appendix B.1), we can apply the standard convergence results for policy evaluation (Sutton & Barto, 2018). The assumption $|A| < \infty$ is required to guarantee that the entropy augmented reward is bounded.

To prove that $\mathcal{T}^{\pi_i^{mix}}$ is a contraction, we need to show that for any $Q_i, Q_i'$,

$$\|\mathcal{T}^{\pi_i^{mix}} Q_i - \mathcal{T}^{\pi_i^{mix}} Q_i'\|_\infty \le \gamma \|Q_i - Q_i'\|_\infty \tag{6}$$

where $\|\cdot\|_\infty$ denotes the infinity-norm on Q-values as $\|Q_i - Q_i'\|_\infty \triangleq \max_{s,a}|Q_i(s,a) - Q_i'(s,a)|$. We begin by evaluating the left-hand side of the inequality:

$$
\begin{aligned}
\|\mathcal{T}^{\pi_i^{mix}} Q_i - \mathcal{T}^{\pi_i^{mix}} Q_i'\|_\infty &= \|(r_{\pi_i} + \gamma \mathbb{E}_{s_{t+1}, a_{t+1}\sim \pi_i^{mix}}[Q_i(s_{t+1}, a_{t+1})]) \\
&\quad - (r_{\pi_i} + \gamma \mathbb{E}_{s_{t+1}, a_{t+1}\sim \pi_i^{mix}}[Q_i'(s_{t+1}, a_{t+1})])\|_\infty \\
&= \gamma \|\mathbb{E}_{s_{t+1}, a_{t+1}\sim \pi_i^{mix}}[Q_i(s_{t+1}, a_{t+1}) - Q_i'(s_{t+1}, a_{t+1})]\|_\infty \\
&\le \gamma \mathbb{E}_{s_{t+1}, a_{t+1}\sim \pi_i^{mix}}[\|Q_i(s_{t+1}, a_{t+1}) - Q_i'(s_{t+1}, a_{t+1})\|_\infty] \\
&\qquad\qquad ( \|\mathbb{E}(x)\|_\infty \le \mathbb{E}\|x\|_\infty \text{ by Jensen's inequality } ) \\
&\le \gamma \|Q_i - Q_i'\|_\infty
\end{aligned}
$$

Since $0 \le \gamma < 1$, this completes the proof that $\mathcal{T}^{\pi_i^{mix}}$ is a contraction mapping. $\qquad\square$

**Theorem G.2** (Mixture Policy Improvement). *Let $\pi_i$ and $Q_i$ be the standard per-task policy and current per-task Q-function for each task $\mathbb{T}_i$. Let $\pi_i^{mix}(s) = \arg\max_{a_j \sim \pi_j(s)\forall j} Q_i(s, a_j)$ be the QMP mixture policy. Define $\mathcal{Q}_i^\pi$ as expected return obtained by rolling out any policy $\pi$. Then, $\mathcal{Q}_i^{\pi_i^{mix}}(s_t, a_t) \ge \mathcal{Q}_i^{\pi_i}(s_t, a_t)$ for all tasks $\mathbb{T}_i = \mathbb{T}_1 \ldots \mathbb{T}_T$ and for all $(s_t, a_t) \in \mathcal{S} \times \mathcal{A}$ with $|\mathcal{A}| < \infty$. Thus, following $\pi_i^{mix}$ guarantees at least as much policy improvement over the soft Q-value $Q_i$ than $\pi_i$.*

*Proof.* Consider the soft Bellman equation for $\pi_i$:

$$
\begin{aligned}
\mathcal{Q}_i^{\pi_i}(s_t, a_t) &= r(s_t, a_t) + \gamma \mathbb{E}_{s_{t+1}\sim p}[V_i^{\pi_i}(s_{t+1})] \\
&= r(s_t, a_t) + \gamma \mathbb{E}_{s_{t+1}\sim p}\left[\mathbb{E}_{a_i \sim \pi_i}[Q_i(s_{t+1}, a_i)]\right] + \gamma \mathbb{E}_{s_{t+1}\sim p}[\mathcal{H}(\pi_i(\cdot|s_{t+1}))] \\
&\le r(s_t, a_t) + \gamma \mathbb{E}_{s_{t+1}\sim p}\left[\mathbb{E}_{a_i\sim\pi_i}\max\left\{\max_{a_j\in\Omega} Q_i(s_{t+1}, a_j), Q_i(s_{t+1}, a_i)\right\}\right] \\
&\quad + \gamma \mathbb{E}_{s_{t+1}\sim p}[\mathcal{H}(\pi_i(\cdot|s_{t+1}))] \qquad\qquad \forall \text{ sets } \Omega \ ( \text{ as } x \le \max\{y, x\}\forall x, y )
\end{aligned}
$$

We choose the set $\Omega$ as samples from all other task policies $a_j \sim \pi_j \forall j = 1...T, j \ne i$

$$
\begin{aligned}
\mathcal{Q}_i^{\pi_i}(s_t, a_t) &\le r(s_t, a_t) + \gamma \mathbb{E}_{s_{t+1}\sim p}\left[\mathbb{E}_{a_i\sim\pi_i}\mathbb{E}_{a_j\sim\pi_j}\max\left\{\max_{a_j} Q_i(s_{t+1}, a_j), Q_i(s_{t+1}, a_i)\right\}\right] \\
&\quad + \gamma \mathbb{E}_{s_{t+1}\sim p}[\mathcal{H}(\pi_i(\cdot|s_{t+1}))] \qquad\qquad\qquad \forall j = 1...T, \forall j \ne i \\
&= r(s_t, a_t) + \gamma \mathbb{E}_{s_{t+1}\sim p}\left[\mathbb{E}_{a_j\sim\pi_j}\max_{a_j} Q_i(s_{t+1}, a_j)\right] + \gamma \mathbb{E}_{s_{t+1}\sim p}[\mathcal{H}(\pi_i(\cdot|s_{t+1}))] \\
&\qquad\qquad\qquad\qquad\qquad\qquad\qquad\qquad\qquad \forall j = 1...T \\
&= r(s_t, a_t) + \gamma \mathbb{E}_{s_{t+1}\sim p}\left[\mathbb{E}_{a_i\sim\pi_i^{mix}} Q_i(s_{t+1}, a_i)\right] + \gamma \mathbb{E}_{s_{t+1}\sim p}[\mathcal{H}(\pi_i(\cdot|s_{t+1}))] \\
&\qquad\qquad (\text{by definition, } \pi_i^{mix}(s) = \arg\max_{a_j\sim\pi_j(s)\forall j} Q_i(s, a_j) ) \\
&= \mathcal{Q}_i^{\pi_i^{mix}}(s_t, a_t)
\end{aligned}
$$

$\qquad\square$

**Why Mixture Policy Improvement works?:** The inequality exists when the actions sampled from other task policies result in better Q-values in the data collected than the task policy $\pi_i$ itself. $\pi_i$ is learned with SAC's policy improvement step to optimize the soft Q-value function. However, in practical execution of the algorithm, $\pi_i$ always lags behind its respective $Q_i$ function which is continuously updated. Therefore, by utilizing action samples from other task policies, Theorem G.2 shows that $\pi_i^{mix}$ can often improve the soft Q-values of the actions that are taken in the environment, but never make it worse with respect to $Q_i$.

**Theorem G.3** (Soft Mixture Policy Iteration and Sample Efficiency Improvement)**.** *Repeated application of (i) soft mixture policy evaluation (Theorem G.1), (ii) soft policy improvement (Lemma 2 from Haarnoja et al. (2018), and (iii) mixture policy improvement (Theorem G.2) to any $\pi_i \in \Pi$ converges to an optimal policy $\pi_i^*$ such that $Q_i^{\pi_i^*}(s_t, a_t) \geq Q_i^{\pi_i}(s_t, a_t)$ for all $\pi_i \in \Pi$ and $(s_t, a_t) \in \mathcal{S} \times \mathcal{A}$ with $|\mathcal{A}| < \infty$.*

*Furthermore, the sample efficiency and rate of convergence is at least as good as SAC in the presence of (iii) mixture policy improvement.*

*Proof.* **Soft Mixture Policy Iteration**: We follow Theorem 1 (Soft Policy Iteration) from Haarnoja et al. (2018) Appendix B.3, which proves that repeated application of soft policy evaluation and soft policy improvement converges to an optimal policy.

Let $\pi_i^k$ be the policy at iteration $k$. By (ii) soft policy improvement lemma, the sequence $Q_i^{\pi_i^k}$ is monotonically increasing. Further, by (iii) mixture policy improvement (Theorem G.2), $Q_i^{\pi_i^{mix;k}}$ also results in a monotonic increase over $Q_i^{\pi_i^k}$. Therefore, the joint application of (ii) and (iii) results in a monotonically increasing sequence of $Q_i^{\pi_i^k}$ at least as fast of a rate than only applying (ii). Since $Q_i^{\pi_i}$ is bounded above for $\pi_i \in \Pi$ (because both the reward and entropy are bounded), the sequence $Q_i^{\pi_i^k}$, the sequence converges to some $\pi_i^*$. The proof to show that $\pi_i^*$ is optimal directly follows Theorem 1 from Haarnoja et al. (2018).

Hence, $\pi_i^*$ is optimal in $\Pi$ and the convergence of Soft Mixture Policy Iteration is at least as fast as Soft Policy Iteration.

$\square$

# H   TEMPORALLY-EXTENDED BEHAVIOR SHARING IN QMP ($H > 1$)

Section G proves that QMP with $H = 1$, i.e., evaluating and rolling out the mixture policy $\pi_i^{mix}$ every step, is guaranteed to converge at least as fast as single-task SAC in tabular settings. This result is also seen empirically (Figure 13). However, this theoretical analysis does not directly extend when $H > 1$. Here, we provide intuition for why temporally-extended behavior sharing is expected to be helpful.

Prior works like Dabney et al. (2020) demonstrate that simply repeating $\epsilon$-greedy's exploratory action in a temporally-extended form results in strong improvements in certain environments. The importance of temporally-extended exploration has also been highlighted in count-based (Osband et al., 2016; Bellemare et al., 2016; Ostrovski et al., 2017; Tang et al., 2017) or curiosity-based (Pathak et al., 2017; Burda et al., 2018) exploration methods, and hierarchical reinforcement learning such as learning options for exploration (Machado et al., 2017; Jinnai et al., 2019b;a; Hansen et al., 2019).

For QMP specifically, we revisit Theorem G.2 where the key reason for QMP's sample efficiency gain is when an inequality exists in the returns obtained from the mixture policy $\pi_i^{mix}$ over the task policy $\pi_i$:

$$\mathcal{Q}_i^{\pi_i^{mix}}(s, a) > \mathcal{Q}_i^{\pi_i}(s, a)$$

This happens when another task policy $\pi_j$ is proposing an action $a_j$ that is better than the task policy $\pi_i$'s action proposal itself. Over a large continuous-valued action space $\mathcal{A}$, $\pi_j$ proposing a good action for Task $\mathbb{T}_i$ by random chance is unlikely. A more likely reason is that $\pi_j$ has been trained on states like $s$ before and its learned behaviors are applicable in Task $\mathbb{T}_i$ as well. This is common

in many task distributions, especially in robotics tasks, because temporally-extended behaviors or skills (Pertsch et al., 2021) are shared across tasks.

Thus, rolling out $\pi_j$ for $H$ steps in the future can, in fact, be better because in the subsequent steps $\pi_j$ can explore more rewarding temporally-extended behaviors beyond the current capabilities of what $Q_i$ can correctly evaluate. Thus, the application of $\pi_j$ in Task $\mathbb{T}_i$ could *shortcut* the need to randomly explore by directly attempting potentially high-rewarding behaviors. While re-estimating $\pi_i^{mix}$ at every time-step and switching between task policies is also a viable strategy as Section G proves, letting $\pi_j$ run for $H > 1$ steps could explore behaviors beyond what the current Q-switch based on $Q_i$ can correctly evaluate.

It is important to note that the strategy of temporally-extended behavior sharing is not expected to help in all task families, and there exist families where QMP with $H = 1$ is the optimal strategy. However, this has been true for temporally-extended exploration works as well (Dabney et al., 2020).

Our key claim is that, for most common task families, such as those observed in robotics and control, temporally-extended behavior sharing is a valid and often significantly outperforming solution. We verify this claim in Figure 13 where we show that both QMP ($H = 1$) and QMP ($H > 1$) always outperform No-QMP (No-Shared-Behavior baseline). However, in tasks like Maze Navigation and Kitchen where the skills or behaviors between tasks have a higher degree of overlap due to other task policies already learning behaviors relevant for the current task, the performance gain of QMP ($H > 1$) is more significant than QMP ($H = 1$). In other tasks like Reacher and Meta-World CDS, QMP($H > 1$) never hurts the performance.

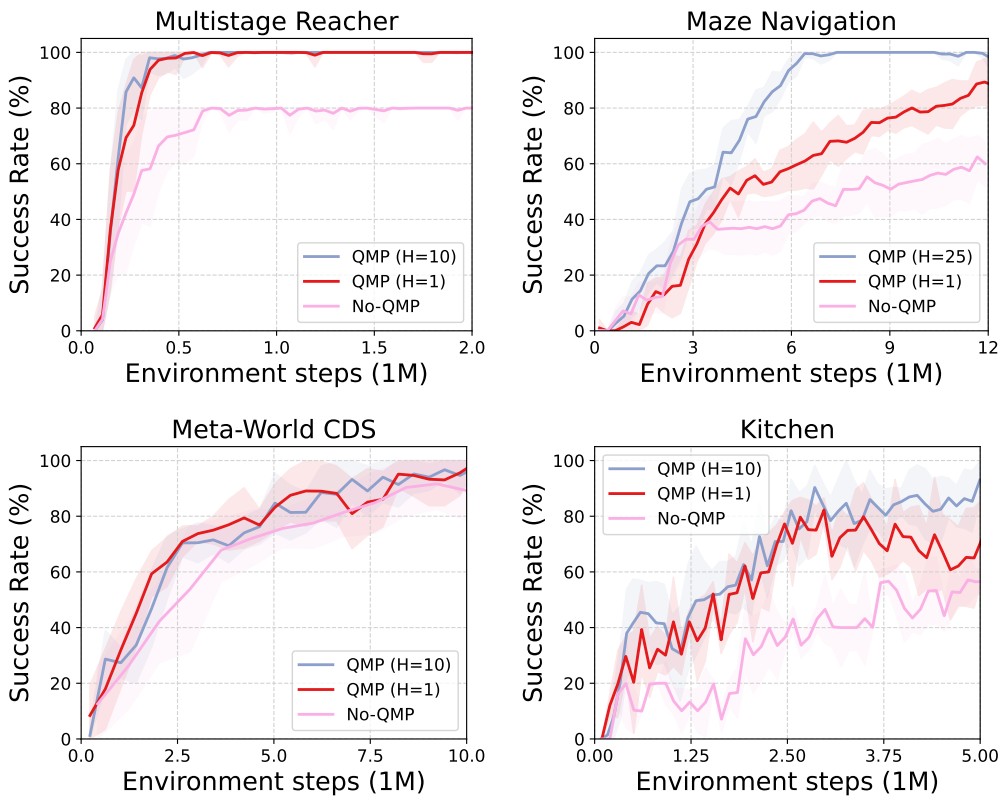

Figure 13: In each case above, QMP with H-step rollouts of the behavioral policy (blue) performs no worse than QMP with 1-step rollouts (red), with the H-step rollouts helping significantly in some tasks. Additionally both versions of QMP outperform the No-QMP baseline.

