# OpenReview forum: "Efficient Multi-task Reinforcement Learning via Selective Behavior Sharing"
_ICLR.cc/2024/Conference — Submitted to ICLR 2024_

### Official Review · Reviewer_H9dR · 2023-10-12

**Soundness:** 2 fair
**Presentation:** 2 fair
**Contribution:** 2 fair
**Rating:** 5
**Confidence:** 4

**Summary:**

This paper advocates for the selective adoption of shareable behaviors across tasks while concurrently mitigating the impact of unshareable behaviors, a proposition that is well-motivated and promising. However, certain sections, notably the abstract and introduction, require further elucidation. More comprehensive conceptual and empirical comparisons with existing literature in this domain are required. Considering the extensive body of relevant work in this field, the evidence presented in the paper falls short of substantiating the acceptance of this work, leading me to recommend a weak rejection.

**Strengths:**

(a) The intuition behind the algorithm design is novel and interesting: using Q-value to identify potentially shareable behaviors and encourage exploration.

(b) The algorithm part (Section 4) is well-presented and easy to follow.

(c) The empirical analysis is detailed and informative about the properties of the proposed algorithm.

**Weaknesses:**

(a) Section 1 requires further elucidation concerning comparisons with prior works and descriptions of the proposed algorithm.

(b) It would be advantageous to include studies on "skills" within related works, given their conceptual similarity to the "shared behaviors" discussed in this paper.

(c) The algorithm's design, which learns a distinct policy for each task, could potentially diminish sample efficiency.

(d) The shareable behaviors can be adopted in a more efficient manner (e.g., forming a hierarchical policy), rather than only used for gathering training data.

(e) The selected baselines for comparisons are kind of weak and can be further strengthened.

(f) The comparisons with baselines depicted in Figure 4 fail to demonstrate notable improvements conferred by the proposed algorithm.

**Questions:**

(a) The definition of " conflicting behaviors" should be elaborated in Section 1.

(b) In Section 2, the authors mention "However, unlike our work, they share behavior uniformly between policies and assume that optimal behaviors are shared across tasks in most states." More explanations are required for "share behavior uniformly" and "assume that optimal behaviors are shared across tasks", where the latter one seems not to be true.

(c) " Yu et al. (2021) uses Q-functions to filter which data should be shared between tasks in a multi-task setting." It would be good to provide more detailed comparisons with this related work.

(d) It should be "argmax" for the equation in Section 3.

(e) Theoretically, the policy network is trained to give actions with the maximized Q-value. That is, $i = \arg\max_jQ_{i}(s, a_j)$ (Line 9 of Algorithm 1) should hold in most cases, which may make this key algorithm design trivial.

(f) The baseline "Fully-Shared-Behaviors" cannot be viewed as a fair comparison, since the agent cannot identify which task it is dealing with. The task identifiers should also be part of the input. There are many research works in this area, such as [1-3] and the ones listed by the authors in Section 2. It would be beneficial to provide comparisons with these works as well.

[1] Sodhani, Shagun, Amy Zhang, and Joelle Pineau. "Multi-task reinforcement learning with context-based representations." In International Conference on Machine Learning, pp. 9767-9779. PMLR, 2021.

[2] Yang, Ruihan, Huazhe Xu, Yi Wu, and Xiaolong Wang. "Multi-task reinforcement learning with soft modularization." Advances in Neural Information Processing Systems 33 (2020): 4767-4777.

[3] Hessel, Matteo, Hubert Soyer, Lasse Espeholt, Wojciech Czarnecki, Simon Schmitt, and Hado Van Hasselt. "Multi-task deep reinforcement learning with popart." In Proceedings of the AAAI Conference on Artificial Intelligence, vol. 33, no. 01, pp. 3796-3803. 2019.

---

> ### Author Response · Authors · 2023-11-23
>
> We thank the reviewer for the constructive feedback especially regarding clarifications in the writing and additional related works to add.  We have incorporated these into the paper.  Additionally, we would like to emphasize that the goal of this paper is to advocate for selective behavior sharing as a promising avenue in multi-task RL which can be complementary to (and not in competition with) many of the existing parameter sharing and data sharing methods that are more well known.  We address your remaining concerns and questions below.
>
> ### Weakness C,D: Distinct policy for each task:
> We clarify that our algorithm’s design is not limited to distinct policies. In fact, the set of experiments in Section 6.2 show that QMP can be implemented with a single multi-head multi-task policy and that the behavior sharing advantages are complementary with parameter sharing advantages.
>
> Furthermore, the choice of distinct v/s shared task policies depends on the task family. Evidently, parameter-sharing is not always optimal and could even hurt training over distinct policies (See Multistage Reacher results in Figure 5 and [3]). Since we consider task sets where different tasks may require conflicting behavior, we base our primary comparisons on distinct policy networks, but also provide results on parameter-sharing + QMP.
>
> ### Weakness E: Weak Baselines
> We believe the reviewer is referring to the comparisons we make in Section 6.1, which are comparisons with other behavior sharing methods and not multi-task RL methods in general.  This supports our argument that QMP is an effective way to share **behaviors** in multi-task RL problems. Importantly, we are not claiming that QMP, or even behavior sharing, is the best MTRL method on its own, and therefore do not compare with complementary non-behavior sharing methods. In fact, we show QMP can be combined with methods like parameter sharing (Section 6.2) to gain the advantages of both methods.
>
> ### Weakness F: Lack of notable improvements
> We would like to note that reviewers QRge, Zv3o, and neqy all noted that the empirical results were comprehensive and demonstrated the effectiveness of QMP.  In our experiments, we chose a wide range of environments and tasks with shared and conflicting behaviors to demonstrate that our simple method of QMP improves sample efficiency and convergence across general multi-task sets, and did not design tasks specifically to favor our method which may have shown more dramatic differences. This is in line with the theoretical proofs in `[Appendix G]` showing that QMP guarantees a weak monotonic improvement over the baseline of no behavior sharing.
>
> ### Question C: Comparisons to data-sharing methods
> - `[Fig 9 (c)]` Yu et al. (2021) [1] requires a ground truth reward function to re-label shared data between methods so it is not applicable to our problem setting. In Fig 9 (c), we show that QMP even outperforms CDS which makes an extra assumption of ground truth reward-relabeling.
> - `[Fig 4]` We do compare against CDS’s follow-up work UDS [2], in Section 6.1, which does not require the true reward labels. QMP consistently outperforms UDS.
>
> ### Question E: Should QMP always choose its own task policy?
> `[Theorem G.2]` While the task policy $\pi_i$ is trained to maximize $Q_i$ over continuous action spaces, it always lags behind $Q_i$, which is itself constantly being updated with new rollouts over training. So the Q-filter can select a different policy if $pi_i$ is not fully optimized for the current state, and a different task policy suggests a better action. When this happens, Theorem G.2 guarantees performance improvement.
>
> ### Question F: “Fully-Shared-Behaviors” baseline
> We use this baseline in section 6.1 specifically to answer “How does QMP compare to other forms of behavior sharing?”, by comparing against a policy that has the same behavior everywhere for all tasks, and therefore mask out the task identifiers to enable policies to learn shared behaviors across tasks.  This is intended as a behavior sharing comparison, not an analysis on parameter sharing in single policies which we do compare against in Section 6.2.
>
> ### [References]
> [1] Yu, T., Kumar, A., Chebotar, Y., Hausman, K., Levine, S., and Finn, C. Conservative data sharing for multi-task offline reinforcement learning. NeurIPS 2021.\
> [2] Yu, T., Kumar, A., Chebotar, Y., Hausman, K., Finn, C., and Levine, S. How to leverage unlabeled data in offline reinforcement learning. ICML 2022.\
> [3] Yu, Tianhe, et al. Gradient surgery for multi-task learning. NeurIPS 2020.\

---

### Official Review · Reviewer_neqy · 2023-10-22

**Soundness:** 2 fair
**Presentation:** 2 fair
**Contribution:** 1 poor
**Rating:** 3
**Confidence:** 4

**Summary:**

This paper introduces Q-Switch Mixture of Policies (QMP) to facilitate the selective sharing of behaviors across tasks, enhancing exploration and information gathering in Multi-task Reinforcement Learning (MTRL). Assuming that different tasks demand distinct optimal behaviors from the same state, QMP employs the Q network of the current task to determine the exploration policy from a pool of all tasks and generate rollout data for efficient training. The method showcases performance improvements across various benchmark scenarios.

**Strengths:**

1. The paper is generally well-written and is easy to follow.
2. The concept of sharing "behavior" instead of data or parameters is intriguing.
3. Empirical results demonstrate that QMP effectively handles multiple tasks with conflicts, and the experiments are detailed and comprehensive.

**Weaknesses:**

1. The central issue with this paper lies in its use of $Q_i$ to evaluate shareable behaviors. In reality, $Q_i(s, a)$ estimates the "expected discounted return of policy $\pi_i$ after executing action $a$ in state $s$," emphasizing that the trajectory to the left is generated by $\pi_i$. However, the authors employ it to evaluate "behavior," which could be interpreted as an action sequence following state $s$." Although the authors acknowledge that "the Q-function could be biased when queried with out-of-distribution actions from other policies," even if we assume that the Q-function fits well, it still struggles to accurately evaluate another policy using only $Q_i(s, \pi_j(s))$.
2. The method, while simple, appears more heuristic in nature and lacks guarantees.
3. The discussion of related works is insufficient and comes across as disjointed and poorly structured.

**Questions:**

Regarding the Weaknesses mentioned, do you think there are ways to address these concerns or clarify the usage of $Q_i$ for evaluating behaviors?

---

> ### Author Response · Authors · 2023-11-23
>
> We thank the reviewer for the helpful feedback on where we can provide additional intuition for our method and on the structure of the related works.  We address your concerns and questions below.
>
> ### Theoretical guarantees
> `[Appendix G]` Please refer to the combined response, where we add convergence and policy improvement proofs for the QMP Algorithm’s modification due to its mixture policy.
>
> ### Using Q-function to evaluate shareable behaviors
> `[Appendix H]` Please refer to the combined response, where we justify (based on prior work, mathematical intuition, and empirical results) why temporally-extended behaviors can effectively obtain high-rewarding trajectories, even when the Q-switch is used to evaluate the mixture policy on the current action.
>
> We note that, for a single action proposal from another task policy, the Q-function can indeed evaluate $Q_i(s, \pi_j(s))$ because $\pi_j(s)$ is just another action to evaluate for $Q_i$. We prove the utility of other tasks in Task $i$’s policy improvement step in Theorem G.2 for H=1, i.e., behaviors of length 1. Further justification for rolling out $\pi_j$ for a sequence of future states is provided in Appendix H.
>
> ### Related Work Discussion
> `[Section 2]` We added several relevant works on MTRL through shared representations and skill learning as a single task behavior sharing example.
>
> We hope the added theoretical results and justifications addresses all your concerns.

---

### Official Review · Reviewer_Zv3o · 2023-10-22

**Soundness:** 2 fair
**Presentation:** 3 good
**Contribution:** 2 fair
**Rating:** 3
**Confidence:** 4

**Summary:**

This paper considers sharing learned behaviors across tasks in multi-task reinforcement learning (MTRL). To preserve optimality, authors propose a method called Q-switch Mixture of Policies (QMP). When training the multi-task policies, QMP estimates the shareability between task policies and incorporates them as temporally extended behaviors to collect training data. Experiments on a wide range of manipulation, locomotion and navigation MTRL task families demonstrate the effectiveness of QMP.

**Strengths:**

1.	This paper is well-written and easy to follow
2.	Sharing learned behaviors among tasks is an interesting and important topic in RL.
3.	Authors conducted extensive experiments and analysis to empirically illustrate the effectiveness of QMP.

**Weaknesses:**

1. Lack of theoretical analysis on the convergence (rate) of QMP.
2. Lack of intuition and detailed analysis on why the proposed method works (i.e., why could QMP make sharing desired behaviors among tasks possible? Please see Question 2 for my concern).

Please refer to my questions below for my concerns.

**Questions:**

1. Why do you choose to roll out H steps instead of just one step after choosing one policy to collect data? QMP just uses the Q function under a particular state to choose the behavioral policy, which cannot guarantee that the selected behavior policy is helpful for the current task after stepping out of the considered state.
2. Will QMP cause undesired behaviors sharing? As I mentioned in Question 1, the selected behavioral policy will roll out for H(>1) steps, which may incurs sub-optimal behaviors.
3. Why choosing the behavioral policy based on the learned Q-function to collect data will essentially share desired behaviors among tasks? I understand that using Q-function as a proxy may help select more better actions. However, the Q-function may be biased and not learned well during training, which could even hurt the learning process.
4. Will QMP even slow down the training process? Say, the learned policy for the current task proposes an action, which is optimal but has an under-estimated Q function. Due to the biased Q function, QMP selects another policy to collect data, which chooses a sub-optimal action. Although the TD update will fix the estimated Q value of the sub-optimal action, it may be more efficient if we directly update the Q value of the optimal action, the thing that we really care about.

I am willing to raise my scores if you could solve my concerns.

---

> ### Author Response · Authors · 2023-11-23
>
> We thank you for your thorough review and your positive comments on the direction of our paper and the extensive empirical results and analysis.  We address your questions and concerns below.
>
> ## Theoretical Analysis on the convergence (rate) of QMP
> `[Appendix G]` Please refer to the combined response, where we add convergence and policy improvement proofs for the QMP Algorithm’s modification due to its mixture policy. We prove that QMP’s mixture policy induces a Bellman operator that is a contraction, and QMP further guarantees a weak monotonic policy improvement over its base algorithm, SAC.
>
> ## Intuition and analysis on why QMP works for behaviors
>
> ### Q1: Using Q-function to evaluate H-step behaviors
> `[Appendix H]` Please refer to the combined response, where we justify (based on prior work, mathematical intuition, and empirical results) why temporally-extended behaviors can effectively obtain high-rewarding trajectories, even when the Q-switch is used to evaluate the mixture policy on the current action.
>
> ### Q2: Undesired Behavior Sharing?
> We address this in detail in the combined response, but in summary, it is possible for QMP to cause undesired behavior sharing through the Q-filter but is able to recover quickly and self-correct due to our method design which only uses other task policies to *gather training data*. We see this in effect in Task 4 of Multistage Reacher which completely conflicts with the other tasks but QMP still performs as well as other baselines that do not share behaviors.
>
> ### Q3: Concerns about bias in learned Q-function
> The reviewer correctly points out that the learned Q-function may be biased, leading to QMP selecting sub-optimal policies. However, this is true for any type of Q-learning where a policy tries to pick actions to maximize a learned Q-function (methods based on SAC [1] and Deterministic Policy Gradient [2]). This generally does not hinder performance in any RL algorithm due to self-correction from online data collection. In fact, many performant Q-learning-based RL algorithms, learn policies directly to maximize a learned Q-function [2, 6, 7].  As explained in Section 4.2 in the paper, the learned Q-function is a critical component of many online RL algorithms [1,2] and has even shown to be a useful filter for high-quality training data [3,4,5]. Our results in conflicting tasks (see QMP vs. No-shared in Figure 8, Task 4) demonstrate that QMP does not suffer from Q-function bias any more than the RL algorithm it is built on top of.
>
> ### Q4: Could QMP slow down the training progress?
> As addressed in Q3, QMP is hindered by the same Q-function bias as most Q-learning algorithms but does not impact sample efficiency in any of the task sets we tried even when there are directly conflicting behaviors (see example in Q2).
>
> We hope the combination of a simple method with strong empirical results and analysis, and the added theoretical results and justifications addresses all your concerns.
>
> ## [References]
> [1] Haarnoja, T., Zhou, A., Abbeel, P., and Levine, S. Soft actor-critic: Off-policy maximum entropy deep reinforcement learning with a stochastic actor. ICML 2018. \
> [2] Lillicrap, T. P., Hunt, J. J., Pritzel, A., Heess, N., Erez, T., Tassa, Y., Silver, D., and Wierstra, D. Continuous control with deep reinforcement learning. arXiv 2015. \
> [3] Yu, T., Kumar, A., Chebotar, Y., Hausman, K., Levine, S., and Finn, C. Conservative data sharing for multi-task offline reinforcement learning. NeurIPS 2021. \
> [4] Nair, A., McGrew, B., Andrychowicz, M., Zaremba, W., and Abbeel, P. Overcoming exploration in reinforcement learning with demonstrations. ICRA 2018. \
> [5] Sasaki, F. and Yamashina, R. Behavioral cloning from noisy demonstrations. ICLR 2020.\
> [6] Silver, David, et al. "Deterministic policy gradient algorithms." ICML 2014.\
> [7] Fujimoto, Scott, Herke Hoof, and David Meger. "Addressing function approximation error in actor-critic methods." ICML 2018.

---

### Official Review · Reviewer_QRge · 2023-11-04

**Soundness:** 2 fair
**Presentation:** 3 good
**Contribution:** 2 fair
**Rating:** 5
**Confidence:** 4

**Summary:**

This paper proposes QMP, a behavior-sharing method in multitask reinforcement learning. QMP uses a mixture of policies to determine which policy is better to collect data for each task. Experiments are conducted on various robotics domains, showing the superior performance of QMP.

**Strengths:**

The idea of this paper is clear and easy to follow.

Experiments are extensive and results are promising, significantly outperforming baselines.

**Weaknesses:**

Some technical details need to be clarified:

1) Since the Q-switch evaluates each $Q_1(s,a_j)$, it inherently assumes all tasks share the same state-action space, or at least assumes the same size of state-action dimension.

2) The reviewer noticed that the paper mentioned 'out-of-distribution action proposal ' in Section 4.3, how does the agent know the action is out-of-distribution? Do you mean the action may be [-10,10] while the action space is [-1,1]?

3) The reviewer is not convinced by the criterion used for collecting data. If the Q-value of some task $j$'s action $a_j$ at state $s$ is the highest, then the QMP will rollout $\pi_j$ for H steps. What if the next state's $Q(s', a_{j'})$ is the worst among all tasks? How does this work, though the results are very promising? When sampling from the dataset, do you filter the samples with lower rewards?

4) How to sample the data from $\pi_i$ and QMP? Is there a specific fraction or equal sampling?

5)

Some questions about experiments:

1) The reviewer feels the comparison is unfair regarding the shared-parameters baseline. The key problem in MTRL is to address the interference when multiple tasks use one network to train the policy. Therefore, a lot of papers come up with different ideas, such as conflict gradient resolution (PcGrad, CAGrad)  and pathway finding (soft modularization, T3S). However, this paper only compared to the basic Multi-head-SAC. While QMP has a designed criterion to select what behaviors to share and how to share. That's why the results in Figure 5 show that 'Parameters + Behaviors' performs worse than 'Behaviors Only'.

The literature review lacks related works such as [1-4].

[1] Conflict-averse gradient descent for multi-task learning

[2] Structure-Aware Transformer Policy for Inhomogeneous Multi-Task Reinforcement Learning

[3] T3S: Improving Multi-Task Reinforcement Learning with Task-Specific Feature Selector and Scheduler

[4] Provable benefit of multitask representation learning in reinforcement learning

**Questions:**

Please refer to the pros and cons part.

---

> ### Author Response · Authors · 2023-11-23
>
> We thank you for your helpful feedback and pointing us to relevant works to include, that we have incorporated a discussion in the revision.  We address your concern about the H-step rollouts in the combined response and the remaining questions below.
>
> ### 1. Shared state-action dimension
> This is a common assumption in many multi-task learning works [1-7]. Experiments on MT10 show that the state-action space does not need to be necessarily shared and only the dimensions are shared. Even specialized architectures like Perceiver that can take multimodal and diverse inputs are complementary to our work, if the Q-function is modeled in that way. So, the requirement of same state-action dimensionality is a consequence of the choice of Q-network, not a limitation of behavior-sharing.
>
> ### 2. Out-of-distribution action proposal
> Out-of-distribution means that the Q-switch, i.e., $Q_i$ may not yet be well-trained on (s, $a_j$), where the $a_j$ is another task’s proposal. In this case, the Q(s, $a_j$) value might be inaccurate. Even if the Q-switch erroneously selects this action $a_j$ to collect data, it would self-correct because by collecting data with $a_j$, the Q-switch would be more accurate on (s, $a_j$) now.
>
> ### 3a. Theoretical grounds for criterion used to collect data
> `[Appendix G]` Please refer to the combined response, where we add convergence and policy improvement proofs for the QMP Algorithm’s modification due to its mixture policy.
>
> ### 3b. Using Q-function to do temporally extended data collection
> `[Appendix H]` Please refer to the combined response, where we justify (based on prior work, mathematical intuition, and empirical results) why temporally-extended behaviors can effectively obtain high-rewarding trajectories, even when the Q-switch is used to evaluate the mixture policy on the current action.
>
> ### 4. Data Sampling Details
> To train the policy and critic for task $i$, we sample uniformly from the data collected in the replay buffer by QMP’s mixture policy $\pi_i^{\text{mix}}$ without needing to do any specialized data balancing or priority sampling based on rewards.  This means that the data collected by QMP is helpful for training $\pi_i$ without additional implementation tricks.
>
> ### Baseline Parameter Sharing Method
> In section 6.2, we use a widely used baseline parameter sharing method, instead of more sophisticated methods like the reviewer points out. This is because this experiment was not meant to compare parameter sharing v/s behavior sharing, but to show that they are **complementary** — which we will clarify in the text. Our goal is *not* to compare parameters-only v/s behaviors-only, because they are orthogonal and complementary ways to share knowledge in MTRL. Our goal is to show that in simultaneous MTRL, behavior-sharing can augment other ways of sharing.
> - Particularly, in Maze Navigation, we see that Parameters-Only already improves significantly over Neither, which shows that methods like PCGrad are not needed in this environment. Our goal is to show that even in this case, Parameters + Behavior sharing further improves significantly over Parameters-Only.
> - As the reviewer points out, in Multistage Reacher, because of the potential conflicts in tasks, parameters-only performs poorly than Neither. However, still Parameters + Behaviors improves over parameters-only. In this case, while combinations of QMP and more sophisticated parameter sharing may yield even better results. However, we believe that the multi-head policy we used is sufficient evidence to demonstrate that QMP is **complementary** with parameter sharing.
>
> We hope this clarifies and addresses the concerns raised.
>
> ### [References]
> [1] Teh et al. Distral. Robust multitask reinforcement learning. NeurIPS, 2017.
> [2] Ghosh et al. Divide-and-conquer reinforcement learning. ICLR, 2018. \
> [3] Yu et al. Conservative data sharing for multi-task offline reinforcement learning. NeurIPS, 2021. \
> [4] Yu et al. How to leverage unlabeled data in offline reinforcement learning. ICML, 2022.
> [5] Kalashnikov et al., Scaling up multi-task robotic reinforcement learning. CoRL 2021b.
> [6] Reed, Scott, et al. "A Generalist Agent." TMLR 2022. \
> [7] Jaegle, Andrew, et al. Perceiver IO: A General Architecture for Structured Inputs & Outputs. ICLR 2021.

---

### Author Response · Authors · 2023-11-23
**Combined Rebuttal**

We thank the reviewers for the constructive feedback. We appreciate the positive notes on the novelty of selective behavior sharing, the clarity and easy-to-follow structure of the paper, and the extensive results and analysis. We address the key concerns below:

## Theoretical guarantees [Zv3o, neqy]

`[Appendix G]` We add convergence and improvement guarantees for QMP in tabular settings. Particularly, we define QMP to follow Soft Mixture Policy Iteration which repeatedly applies 3 components:
1. **Soft Mixture Policy Evaluation**: Theorem G.1 proves that the Bellman backup operator resulting from application of QMP has convergence guarantees, by proving that it is a contraction.
2. **Soft Policy Improvement**: This is the update rule of Soft Actor-Critic and proven to guarantee policy improvement in Lemma 2 of Haarnoja et. al (2018).
3. **Soft Mixture Policy Improvement**: Theorem G.2 proves that QMP’s mixture policy guarantees a weak monotonic improvement over the standard policy. Specifically, taking the maximum over multiple policies better optimizes the soft Q-value function $Q_i$.  In MTRL, other task policies could sometimes be better at maximizing $Q_i$, as they might have already acquired the behavior currently being learned in Task i.

Finally, Theorem G.3 (Soft Mixture Policy Iteration) shows:
- convergence: repeated application of these 3 steps in QMP converges to an optimal policy.
- sample efficiency: due to the Soft Mixture Policy Improvement from Theorem G.2.

In addition to these theoretical guarantees, we believe the combination of a simple and intuitive method with strong empirical results and analysis is a valuable contribution to multi-task RL research, especially in taking the relatively under-explored approach of behavior sharing.

## Justification for Temporally-Extended Behavior Sharing [QRge, Zv3o, neqy]

`[Appendix H, Figure 13]` As the reviewers point out, in temporally-extended QMP, it is true that the Q-filter selects a policy by evaluating only the next action, so we are not choosing the optimal policy over the next H steps that we roll out. We justify why temporally-extended behavior sharing works in Appendix H and Figure 13:
- **Mathematical Intuition**: When QMP’s Q-switch for Task $i$ selects another task’s policy $\pi_j$’s action proposal, this likely means that $\pi_j$ has been trained on the current state and its learned behaviors are applicable in Task $i$ as well. Thus, rolling out $\pi_j$ for H steps in the future could directly obtain high-rewarding trajectories, even when $Q_i$ is not prepared yet to evaluate those future states as the agent in Task $i$ has not yet explored those states at all. This *shortcuts* the need to randomly explore and stumble upon good behaviors.
- **Empirical Evidence**: Our key claim is that, for most common task families, such as those observed in robotics and control, QMP with H>1 is valid and often a significantly outperforming solution. Figure 13 verifies this claim by showing that both QMP (H=1) and QMP (H>1) always outperform No-QMP, but, in tasks like Maze Navigation and Kitchen QMP (H>1) outperforms QMP (H=1). Also, in other tasks like Reacher and Meta-World CDS, QMP(H>1) never hurts the performance of QMP (H=1), even though the Q-filter only evaluates the next proposed action.
- **Prior Work**: Dabney et. al (2020) show that temporally-extended $\epsilon$-greedy outperforms naive $\epsilon$-greedy in certain tasks. There is also evidence of temporally-extended exploration in count-based and curiosity-based exploration, and options learning in hierarchical reinforcement learning.



## Undesired Behavior Sharing [Zv3o, neqy]

It is possible for QMP to cause undesired behavior sharing through the Q-filter. However, QMP is able to recover quickly in the many multi-task setups with conflicting behaviors we have explored, and does not hurt learning performance.  We can see this by looking at QMP’s performance in Task 4 of Multistage Reacher where any behavior sharing would be unhelpful.  QMP quickly learns to not share behaviors from other policies (Appendix Figure 11) and performs as well as any behavior sharing baseline (Appendix Figure 8) in this task.

QMP’s ability to recover from undesired behavior sharing stems from the method design which only uses other task policies to gather training data and not to train the task policy directly.  When undesired behaviors are tried, QMP ensures that these behaviors only modify the data collection and not the policy training. Theorem G.2 also proves that QMP is at least as good as your own task policy when rolled out for one step.

We thank the reviewers for their suggestions that helped strengthen our paper by providing theoretical guarantees and a concrete justification of why temporally-extended behavior sharing works well. We hope our changes, when seen together with the extensive empirical results of QMP, resolve all concerns of the reviewers.

---

### Meta-Review · Area_Chair_ZjhZ · 2023-12-04

**Metareview:**

This paper introduces a novel, generally applicable behavior-sharing formulation that selectively leverages other task policies as the current task's behavioral policy for data collection.

**Reviewers have reported the following strengths:**

- Novelty and significance;
- Experimental results.

**Reviewers have reported the following weaknesses:**

- Lack of motivation;
- Lack of theoretical guarantees.

**Decision**

This paper received negative reviews, mostly pointing out the lack of guarantees of the described method. I recommend the authors improve the paper in this direction.

**Justification For Why Not Higher Score:**

N/A

**Justification For Why Not Lower Score:**

N/A

---

### Decision · Program_Chairs · 2024-01-16

Reject